# Push-Out Test and Hysteretic Performance Study of Semi-Rigid Shear Keys with the Triple-Folded Web of Flange

Zhenshan Wang [1,2,*], Huaqian Qin [2], Yong Yang [2], Yunhe Liu [2], Hongchao Guo [2] and Hongchen Wang [3]

1 State Key Laboratory of Northwest Arid Ecological Water Conservancy Engineering, Xi'an University of Technology, Xi'an 710048, China
2 School of Civil and Architectural Engineering, Xi'an University of Technology, Xi'an 710048, China; a58391390@163.com (H.Q.); yy17829076947@163.com (Y.Y.); liuyhe@xaut.edu.cn (Y.L.); ghc-1209@163.com (H.G.)
3 China Northwest Architecture Design and Research Institute Co., Ltd., Xi'an 710048, China; lwf201977@163.com
* Correspondence: wangzhenshan@xaut.edu.cn

**Abstract:** The PBL (Perfobond Leiste) shear connector has the advantages of high bearing capacity and strong constraint ability; however, the traditional PBL shear connector has strong and weak axis problems, and its stiffness is large, resulting in weak deformation ability. To this end, this paper proposes a new type of flange triple-web shear key and obtains the new shear key's mechanical properties and failure mechanism through the push-out test. The results show that the failure mode of the new shear key is the deformation of the steel plate on the web and the edge of the opening, which has a high bearing capacity, outstanding deformation ability, and good integrity with concrete, showing obvious semi-rigid characteristics. Through numerical analysis, the effects of flange width, web height, and steel plate thickness on the mechanical properties of shear keys are obtained. Based on the fitting analysis method, the calculation formula of shear key bearing capacity is proposed. Finally, the horizontal seismic performance of the shear key is numerically simulated. It is found that the hysteretic curve of the shear key is full and shows good energy dissipation capacity.

**Keywords:** semi-rigid shear key; push-out test; numerical simulation; calculation of bearing capacity; seismic performance

## 1. Introduction

Steel–concrete composite structure has the advantages of high strength and convenient construction, which has been widely used in structural engineering [1,2]. Reliable shear connectors are the key components to ensure the performance of composite structures. Their main functions are to transfer the longitudinal shear force and resist the longitudinal separation between steel and concrete slabs [3,4]. As a flexible shear member, the stud has mature technology, a wide application range, strong deformation ability, and convenient welding. Its bearing capacity increases with the increase of stud diameter and yield strength [5,6]. However, the bolt cannot be effectively removed and reused at the end of service life. Therefore, the high-strength stud connector is used to replace the stud to study its shear performance. It is found that the shear performance of high-strength bolt connectors mainly depends on the bolt diameter, tensile strength, and concrete strength, which is less affected by the preload [7,8]. However, on the whole, the stiffness of the stud and high-strength stud connectors is small, the bearing capacity is low, and the relative slip is large.

For engineering with strict deformation control, rigid shear connectors, such as angle steel and T-shaped plate connectors, are often used, which can not only meet the high bearing capacity but also effectively limit the slip of components. However, the stiffness of such connectors is large, and the deformation and energy consumption under earthquakes

are poor, resulting in brittle failure of concrete at the contact position of shear keys [9–15]. In order to further improve the connection performance between steel and concrete, the PBL shear bond is proposed by scholars. It is found that increasing the opening area and the thickness of the steel plate can improve the bearing capacity and ductility of the members [16–19]; setting flanges can significantly improve the bearing capacity of shear keys and increase the constraint range of concrete slabs [20–22]; inserting steel bars at the opening can improve the bearing capacity of shear keys, and the deformation capacity is also significantly improved [23–25]. Under external pressure, the shear capacity increased slightly [26]. Although such shear keys have good mechanical properties, the primary purpose of most studies is to improve the strength of shear keys, and the matching of strength and stiffness is not studied in depth. Based on PBL shear keys, some scholars proposed embedded shear keys and found that the stiffness of embedded shear keys was small, and the fatigue resistance was relatively weak. With the maturity of technology, the structural characteristics of embedded shear keys have been continuously improved. It is found that the diameter of steel plate openings and the strength of concrete are the main factors affecting the ultimate bearing capacity of such shear keys. Moreover, the diameter and strength of penetrating steel bars and the thickness of corrugated steel plates have relatively small effects on the bearing capacity [27–29]. El-Zohairy et al. [30–32] carried out fatigue performance tests on composite beams with stud connectors. They mainly studied the effect of the number and arrangement of studs on the fatigue behavior of composite beams. According to the research on the mechanical properties of PBL shear keys under earthquake action, it is found that the stress forms under static and reciprocating actions are significantly different. The bearing capacity of shear keys is significantly reduced, and the stiffness also has a certain degree of degradation [33–37].

Many scholars in China and abroad have studied the new connection forms in recent years. Compared with the traditional form, the mechanical properties of the Y-type shear connector are improved in shear strength and stiffness [38–40]. Nodir et al. [41] studied the pull-out form of the composite structure of PBL connectors wrapped with CFRP (carbon fiber reinforced polymer). The results showed that under the constraint of CFRP, PBL connectors realized the organic combination of deformation and bearing capacity. At the same time, Fan et al. [42] proposed a PBH (perfobond hoop) connector. They found that the PBH connector has better mechanical properties and higher ultimate bearing capacity than the PBL connector.

Currently, the research on PBL shear keys mainly focuses on their static changes, and the performance degradation law and damage mechanism under earthquake action lack clear understanding. In addition, the traditional PBL shear key has strong and weak axis problems, and it is difficult to give full play to the mechanical properties of the shear key for the seismic action from the weak axis direction. As a kind of rigid shear key, the same problem of poor deformation capacity exists, which leads to the damage of concrete at the contact position of the shear key. This paper proposes a flange triple-folded web semi-rigid shear connector based on the above analysis. An open-hole folding web design can solve the problem of strong and weak axes and increase the shear key and concrete embedded action to achieve the effective unification of bearing capacity and deformation. This paper obtains the damage mode, bond-slip ultimate bearing capacity, and deformation capacity of this new shear key using the push-out test method. The damage mode and force mechanism are given. Using the parametric analysis method studies critical factors such as shear key steel plate thickness, web height, opening diameter, and the design suggestions and ultimate bearing capacity calculation equations. At the same time, the seismic action analysis of this new shear key was carried out to obtain the load-displacement hysteresis performance and compare and analyze the load-carrying capacity and stiffness degradation under the static force and hysteresis action. The research can reveal the force behavior and changes rule of the new shear key under static and seismic effects, as well as fill the relevant gaps, and the research results can provide the basis for the engineering application of the new shear key.

## 2. Experimental Program

### 2.1. Test Specimens

Design concept: the web adopts the form of three folded plates, using the deformation characteristics of the folded structure to improve the ductility of members; at the same time, the stiffness of the two main axes can be balanced to avoid the problem of strong and weak axes, which is more conducive to resisting earthquake effects in different directions. Web openings: can improve the embedded effect with concrete, and shear keys can form a bite effect with concrete to improve the cooperative work ability of the two. Flange: can enhance the stability of the web and anti-lift effect. Specific size: stud M22 × 90 mm, shear key web and flange thickness of 6 mm, web length of 68 mm, height of 90 mm, a middle of the web diameter of 25 mm, height of 70 mm long circular hole, and specific size as shown in Figure 1a. On the amount of steel, a shear key is approximately equal to two studs, and the comparative test is designed according to this, as shown in Figure 1b.

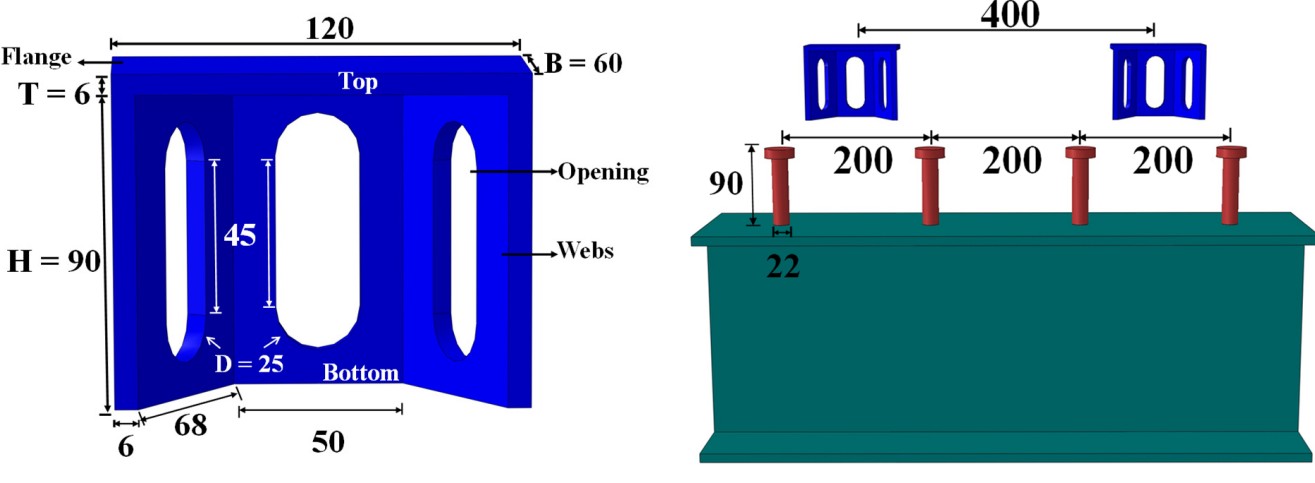

(**a**) Shear key with triple-folded web of flange          (**b**) Comparative experiment design

**Figure 1.** Distribution and size of shear keys.

In this paper, four groups of specimens are designed. The parameters are listed in Table 1. The composition and forming process of the specimens are shown in Figure 2. The specific process is as follows: (1) welding the bottom of shear keys to the upper flange surfaces of steel beams and precast 60 mm thick reinforced concrete slab, assembled on both sides of the steel beam; (2) the joint (shear key width) is reserved to fill the joint; and (3) layout of surface steel overall pouring 70 mm concrete. The slab is 580 mm × 550 mm × 130 mm, using C30 concrete, of which 60 mm is precast slab and 70 mm post-cast. The longitudinal and transverse rebars in the plate are HRB400 rebars with a diameter of 8 mm, and the rebars in the hole are HRB400 rebars with a diameter of 12 mm, and the intermediate steel beam is H-shaped steel Q235 grade. The specification is HW300 mm × 200 mm × 8 mm × 12 mm.

**Table 1.** The number and form of specimens.

| Specimen Number | Form | Penetrated Steel Bar |
|---|---|---|
| S-1 | Triple-folded plate shear key | No |
| S-2 | Triple-folded plate shear key | No |
| S-3 | Triple-folded plate shear key | Yes |
| S-4 | Stud | — |

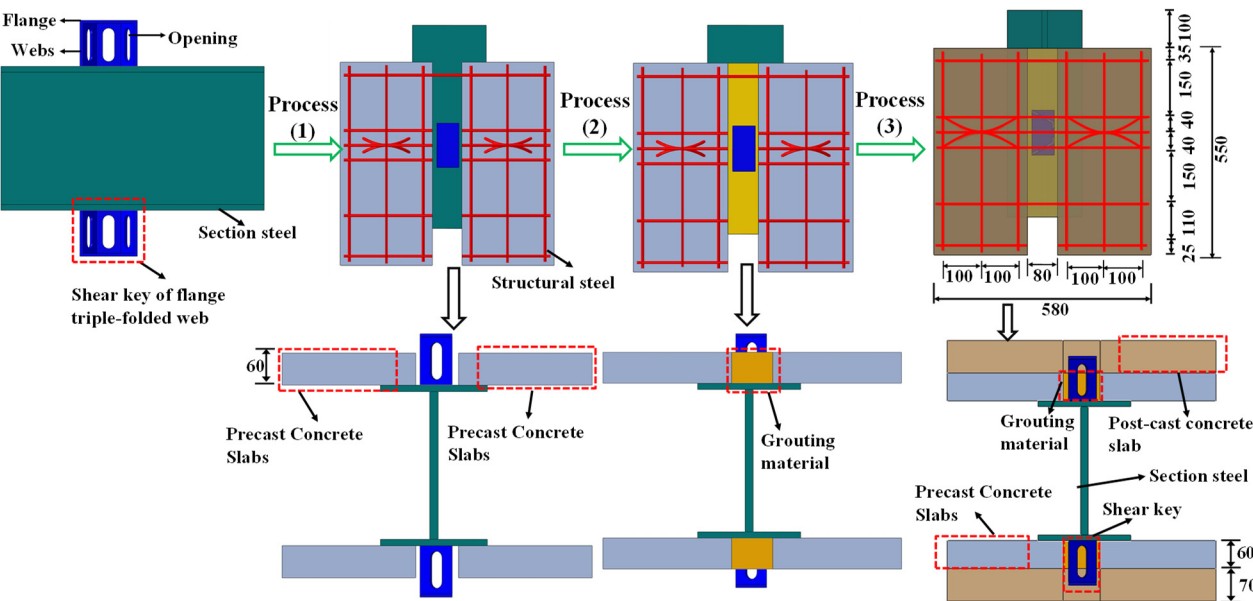

**Figure 2.** Composition and formation of the specimen.

### 2.2. Material Properties

The material properties were measured for the concrete, steel plate, and steel bar used in the specimen, and the specific results are shown in Tables 2 and 3.

**Table 2.** Test results of material properties of concrete.

| Material Type | $f_{ck}$ (MPa) | $f_{cu,k}$ (MPa) | $f_c$ (MPa) | $E_c$ (MPa) |
|---|---|---|---|---|
| Commercial concrete | 27.0 | 40.4 | 19.3 | 33,827.1 |

NOTE: $f_{cu,k}$: standard value of concrete cube compressive strength; $f_{ck}$: axial compression strength of concrete; $f_c$: designed axial compressive strength of concrete; $E_c$: modulus of elasticity of concrete.

**Table 3.** Test results of steel plate and reinforcement.

| Specimen Number | $f_y$ (MPa) | $f_u$ (MPa) | $\varepsilon_y$ (%) | $\varepsilon_u$ (%) | $E$ (GPa) |
|---|---|---|---|---|---|
| Reinforcement (Φ6) | 572.0 | 642.0 | 0.5 | 3.2 | 189.0 |
| Reinforcement (Φ8) | 472.0 | 662.0 | 0.5 | 6.3 | 183.0 |
| Reinforcement (Φ12) | 509.0 | 583.0 | 0.3 | 9.9 | 187.0 |
| Steel plate | 255.0 | 410.0 | 0.8 | 9.5 | 214.0 |

NOTE: $f_y$: yield strength; $f_u$: tensile strength; $\varepsilon_y$: yield strain; $\varepsilon_u$: tensile strain; $E$: elastic modulus of steel.

### 2.3. Test Setup and Loading Procedure

The loading device uses a long column press (Figure 3), and the measuring point D (Figure 3c) is arranged on the steel beam web to measure the slip of the steel beam. The pressure is obtained by the pressure sensor of the long column press. The loading system adopts force-displacement mixed loading, and the steel beam is controlled by force before sliding. The loading rate is 0.15 kN/s, and each loading is 100 kN for two minutes. Displacement control was adopted after sliding, and the loading rate was 0.03 mm/min. When the bearing capacity decreased to 85% of the peak load, the loading was stopped.

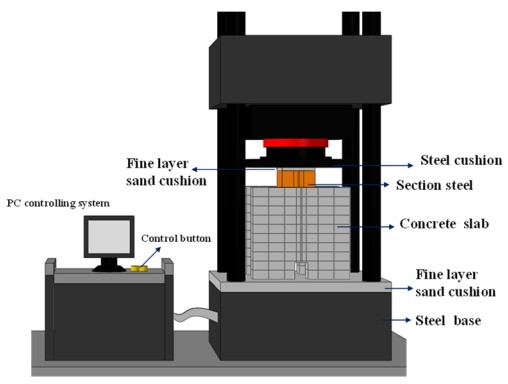

(**a**) Loading device schematic

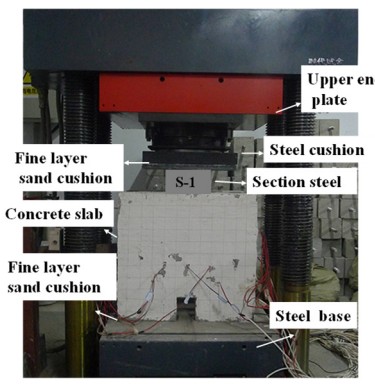

(**b**) Actual situation of loading device

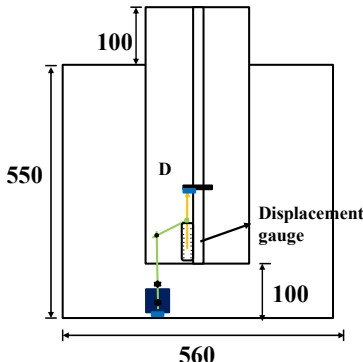

(**c**) Layout of slip measuring points

**Figure 3.** Experiment loading and testing.

## 3. Test Results and Discussion

### 3.1. Failure Modes

The crack development process of S-1, S-2, and S-3 specimens is the same, as shown in Figure 4a–c. There is no obvious phenomenon at the beginning of loading; when the load reached 250 kN, the steel beam began to slip, and the bottom of the concrete slab first appeared with vertical cracks. As the load continues to load, cracks continue to extend upward; when the load reaches the ultimate load, 45° oblique cracks appear on both sides of the shear key, and new vertical cracks appear at the bottom of the concrete slab. The bearing capacity decreases to 85% of the ultimate bearing capacity, stopping the loading. The cracks develop around the concrete slab, and finally, the trapezoidal main cracking area is formed. Among them, the S-3 specimen is inserted with steel bars at the opening, the concrete slab is damaged more seriously, and the crack distribution is larger and more uniform. When the concrete is stripped, the failure of the shear key is shown in Figure 4a–c. The shear key bends overall along the push-out direction, and the lateral web and the opening edge show apparent buckling.

The S-4 specimen is a stud shear connector, and there is no apparent phenomenon at the beginning of loading. When the load reaches 150 kN, the steel beam begins to slip, and vertical cracks appear at the stud position. When the ultimate load is reached, no new cracks are found in the front of the concrete, and oblique cracks appear at the bottom of the side. When the bearing capacity decreases to 85% of the ultimate bearing capacity, the loading is stopped. Overall, under the constraint of stud connectors, the concrete composite plate has delamination, and the specimen's integrity is poor. When the concrete is stripped, the failure of the stud is shown in Figure 4d. The apparent bending occurs, and the deformation in the middle of the stud is the most serious.

The push-out test found that the failure process and mode of the concrete slab are roughly similar. First, vertical, oblique cracks appear at the concrete slab's bottom. Furthermore, cracks are formed in the oblique direction of the shear key and gradually develop in the direction of 45° around. Subsequently, the vertical concrete plate along the shear key cracked; finally, a trapezoidal cracking zone centered on shear keys is formed, with the top cracking reaching about 1/3 plate length, the bottom about 2/3 plate length, and the height of 7/10 plate height (as shown in Figure 5a). The failure modes of shear keys are basically the same. Along the push-out direction, obvious overall deformation occurs, and apparent buckling occurs on both sides of the steel plate. The edge of the opening has different degrees of deformation. The specific failure mode is shown in Figure 5b. The traditional stud concrete plate is only slightly damaged, the junction of the composite plate has a layered trend, and the bending deformation occurs in the middle of the stud. From the perspective of failure mode, the new shear key has a more extensive constraint range, which can significantly improve the integrity of the concrete composite slab.

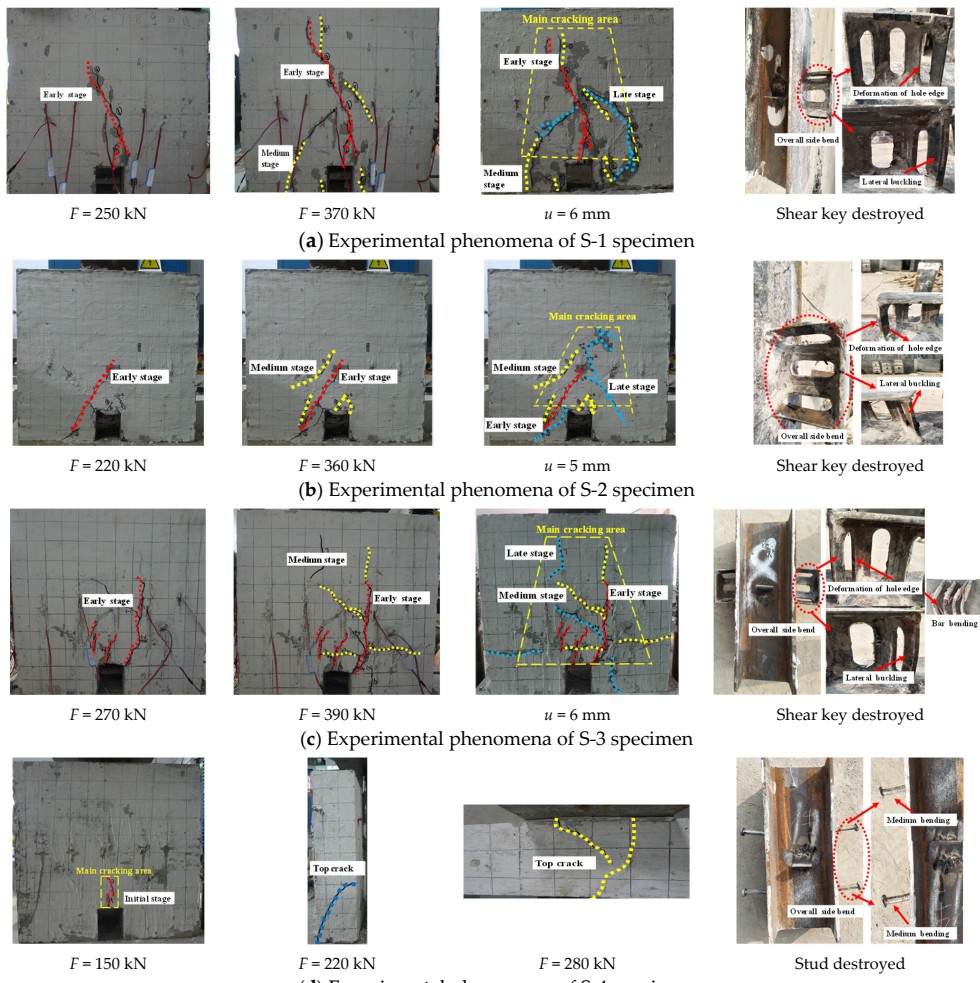

**Figure 4.** Experimental phenomena.

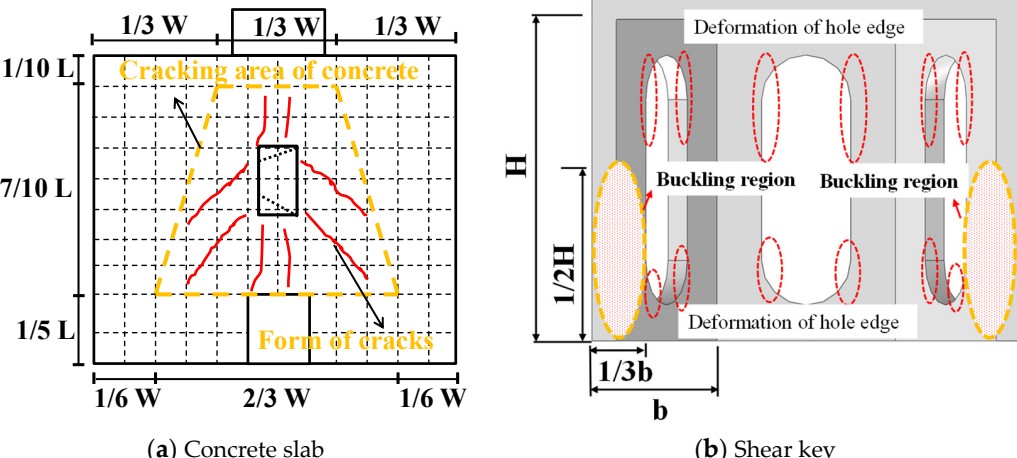

(**a**) Concrete slab　　　　　　　　　　　　　　　　　(**b**) Shear key

**Figure 5.** The failure mode of shear keys with the triple-folded web of flange.

### 3.2. Load-Slippage Curve

　　　The load-displacement curves analysis of each specimen is shown in Figure 6. The test process is divided into four stages: bonding, sliding, strengthening, and failure. When the steel beam slips, the triple-folded shear key has an obvious strengthening effect, and the bearing capacity increases rapidly while the bearing capacity of the stud increases slowly. After reaching the peak load, the stiffness degradation, bearing capacity decline, concrete

slab failure, the traditional stud stiffness, and strength degradation are slow. The shear key with triple-folded web also showed good deformation ability, and the decrease in bearing capacity was relatively slow. The effect of inserting steel bars at the opening on the bearing capacity of the specimen is small, but after loading to the peak load, the bearing capacity and stiffness decrease slowly, and the deformation capacity is significantly improved.

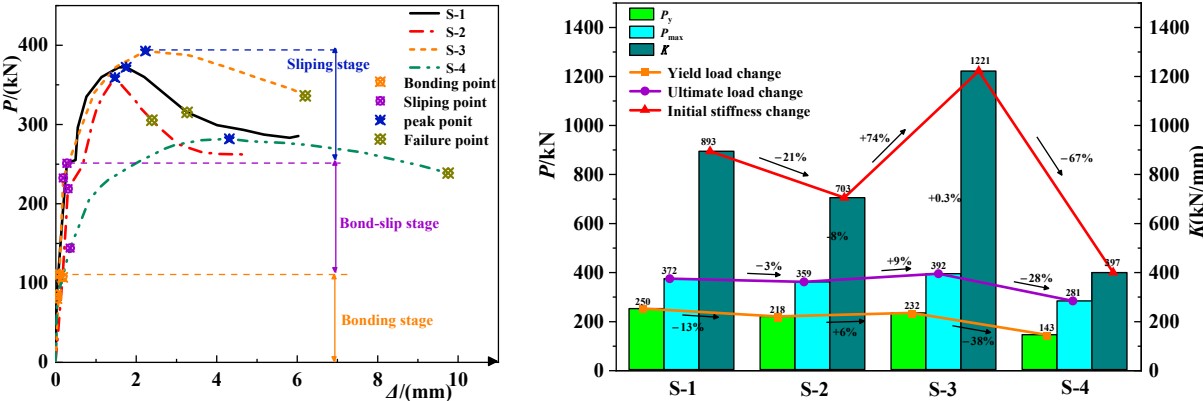

**Figure 6.** Analysis of load-displacement curves.

The characteristic values of load and displacement are shown in Table 4. Compared with studs, the slip load of shear keys increases by more than 45%, the ultimate bearing capacity increases by more than 32%, and the failure displacement reach 55% of studs. Compared with the S-1 and S-2 specimens, the specimens' ultimate load with steel bars inserted at the opening increased by 6%. The bearing capacity margin was basically the same. In summary, the bearing capacity of the triple-folded shear key is significantly higher than that of the stud. It has good embedded performance and stronger integrity with the concrete slab. The effect of steel bars through the opening on the bearing capacity is small, but the stiffness degradation of the specimen is slow, and the ductility performance is good.

**Table 4.** Eigenvalues of load and displacement.

| Specimen | Slip Load $P_s$/kN | Ultimate Load $P_{max}$/kN | Displacement of Ultimate Load $\Delta_{max}$/mm | Failure Load $P_u$/kN | Displacement at Failure Load $\Delta_u$/mm | Bearing Capacity Margin $P_{max}/P_y$ |
|---|---|---|---|---|---|---|
| S-1 | 250 | 372 | 1.8 | 284 | 3.3 | 1.5 |
| S-2 | 218 | 359 | 1.5 | 304 | 2.4 | 1.6 |
| S-3 | 232 | 392 | 2.2 | 335 | 6.2 | 1.7 |
| S-4 | 150 | 281 | 4.0 | 238 | 9.8 | 1.9 |

NOTE: Sliding load $P_s$ is the load corresponding to steel sliding; ultimate load $P_{max}$ is the peak load when slipping; failure load $P_u$ is the load corresponding to a decrease in load to 85% of the peak load.

## 4. Finite Element Simulation of Test Specimens

### 4.1. Finite Element Model

The finite element model is established by ABAQUS 6.14, as shown in Figure 7. Solid element is suitable for large deformation analysis, and the calculation accuracy is high. When the mesh has bending deformation, the analysis accuracy will not be significantly affected. Therefore, concrete slabs, H-beams, concrete teeth, and shear keys adopted solid element C3D8R (8-node 6-plane linear reduction integral element); the slenderness ratio of steel bars is large, so the bending, shear, and torsion of steel bars are ignored, and the tension effect is only considered. Therefore, the diffraction frame element T3D2 (2-node linear element) is used. Tie constraints are used to simulate the contact between studs and steel beams, shear keys and steel beams, concrete blocks and concrete slabs, and tie steel bar and concrete slabs. The remaining constraints are face–face contact, normal hard

contact, tangential penalty coefficient, friction coefficient is 0.3, and the tangential penalty coefficient between the stud and the surrounding concrete is 0.25. The upper and lower surfaces of the stud head are contacted with the concrete without friction in the tangential direction. Due to the smooth surface of steel beam and concrete slab, when setting contact properties, the tangential behavior adopts a penalty coefficient, the friction coefficient is 0.05, and the normal behavior adopts hard contact, allowing separation after contact.

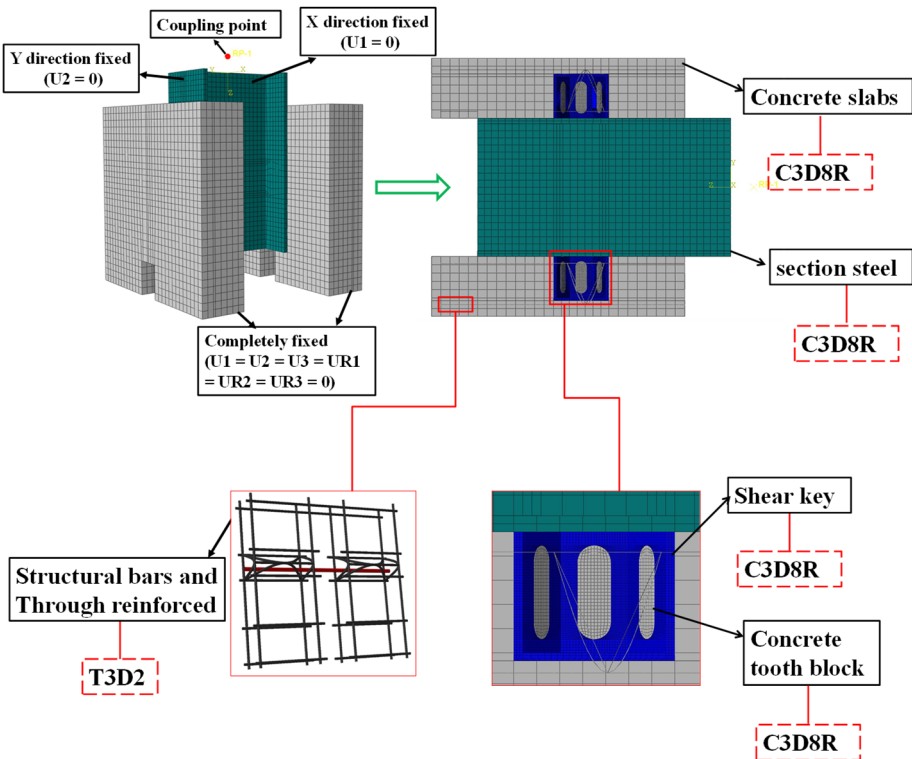

**Figure 7.** Model establishment.

The overall unit size of the concrete slab and steel beam is 25 mm, and the unit size of the concrete tooth block is 3 mm; the shear key unit size is 5 mm, and the steel plate is divided into three layers according to the thickness direction, and the grid of key parts is encrypted. The bottom of the concrete slab is in completely fixed boundary conditions (U1 = U2 = U3 = UR1 = UR2 = UR3 = 0); to avoid the buckling of the steel beam during loading, the displacement/rotation angle is set to 0 in the X direction, and Y direction (U1 = 0, U2 = 0); the reference point 1 (RP-1) 50 mm away from the top center of the steel beam is selected as the control point of the coupling constraint. The displacement load of 6 mm is applied to the positive direction of Z. To improve the convergence, when meshing, the grid quality is improved by adjusting the grid size and attribute. When the contact master-slave surface is selected, the surface with large stiffness and coarse mesh is selected as the main surface; when concrete tooth blocks contact with the shear key, choose "delete interference" to remove the gap between the two.

### 4.2. Material Constitutive Relationship

Reinforcement and steel use the trilinear constitutive model and bilinear kinematic hardening model, as shown in Figure 8. Under monotonic loading, the linear slope of the elastic stage is elastic modulus. After reaching the yield strain $\varepsilon_y$, it enters the yield stage until the strain reaches $10\varepsilon_y$. Then it enters the strengthening stage until the stress and strain reach the ultimate stress $f_u$ and ultimate strain $\varepsilon_u$. Under the action of reciprocating load, it is divided into the elastic stage and strengthening stage. The elastic modulus of the steel plate in the elastic stage is E = $2.14 \times 10^5$ MPa, and the steel bar is E = $1.89 \times 10^5$ MPa; Poisson's ratio $\mu = 0.3$, and strengthening modulus E' = 0.01E.

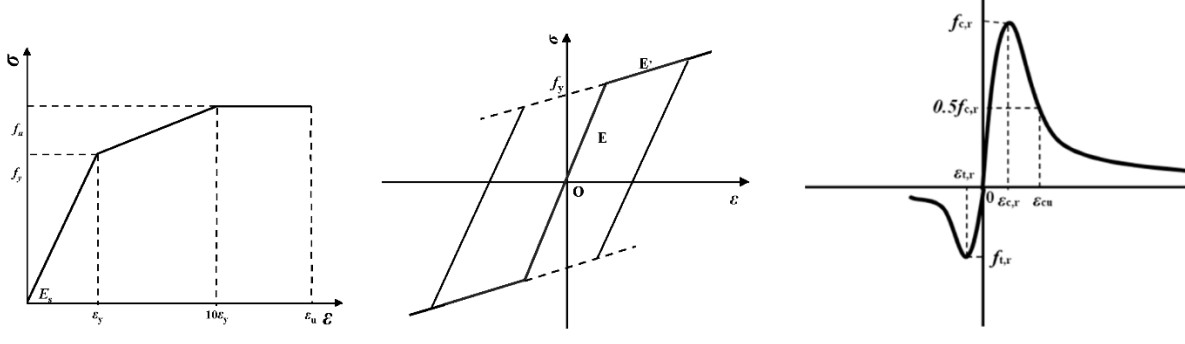

**(a)** Constitutive model of steel under monotonic loading　　**(b)** constitutive model of steel under cyclic loading　　**(c)** Constitutive model of concrete

**Figure 8.** The material constitutive.

A plastic damage (CPD) model is adopted for concrete; the uniaxial stress-strain curve adopts the constitutive relationship recommended in *GB50010-2010* [43], shown in Figure 8c; according to the results of material testing, where the elastic modulus of concrete $E_c = 33.83 \times 10^3$ MPa, and the Poisson's ratio is 0.2, it can be determined by the following formulae that:

$$\sigma_t = (1 - d_t) E_c \varepsilon \tag{1}$$

$$\sigma_c = (1 - d_c) E_c \varepsilon \tag{2}$$

where $\sigma_t$ is the tensile stress on the concrete; $\sigma_c$ denotes compressive stress on the concrete; $d_t$ is the uniaxial tensile damage evolution parameter of concrete; $E_c$ is the elastic modulus of concrete; $\varepsilon$ is the strain in the concrete; and $d_c$ is the uniaxial compressive damage evolution parameter of concrete.

When tensile stress is applied:

$$d_t = \begin{cases} 1 - \rho\left[1.2 - 0.2x^5\right] & x \leq 1 \\ 1 - \dfrac{\rho_t}{\alpha_t (x-1)^{1.7} + x} & x > 1 \end{cases} \tag{3}$$

When compressive stress is applied:

$$d_c = \begin{cases} 1 - \dfrac{\rho_c n}{n - 1 + x^n} & x \leq 1 \\ 1 - \dfrac{\rho_c}{\alpha_c (x-1)^2 + x} & x > 1 \end{cases} \tag{4}$$

where $\rho_c = \dfrac{f_{c,r}}{E_c \varepsilon_{c,r}}$, $x = \dfrac{\varepsilon}{\varepsilon_{c,r}}$, $n = \dfrac{E_c \varepsilon_{c,r}}{E_c \varepsilon_{c,r} - f_{c,r}}$, $x = \dfrac{\varepsilon}{\varepsilon_{t,r}}$, $\rho_t = \dfrac{f_{t,r}}{E_c \varepsilon_{t,r}}$, and $\alpha_c$ is 1.23 for the parameter of the compression descending section; $\alpha_t$ is 1.95 for the parameter of the downward section under tension; $f_{c,r}$ represents the representative value of concrete compressive strength, according to the test results of material properties (27.6 MPa); $f_{t,r}$ is the representative value of concrete tensile strength (2.5 MPa); $\varepsilon_{c,r}$ is the compressive strain at failure (0.00016); $\varepsilon_{t,r}$ is the tensile strain at failure (0.00011); and $\varepsilon_u$ is the compressive strain of concrete when the stress in the descending section is $0.5f_{c,r}$. The plastic damage factor is calculated through the following calculation:

$$d = 1 - \sqrt{\dfrac{\sigma}{E_0 \varepsilon}} \tag{5}$$

where $d$ is the plastic damage factor of the CPD model; and $E_0$ refers to the initial elastic modulus of concrete.

### 4.3. Finite Element Results

The studs' simulation results are consistent with the experimental phenomena, shown in Figure 9. The maximum stress appears in the middle of the studs, and the lateral bending occurs along the push-out direction. The concrete slab is a dot-like constraint, and the constraint range is small. The comparison of load-displacement curves is shown in Figure 9d. Before sliding, the curves basically overlap; as the load increases and the curves enter the sliding stage. Overall, the variation trend of load-displacement curves is basically the same, and the numerical calculation bearing capacity is slightly higher, reaching 305 kN, with an error of about 8.5%.

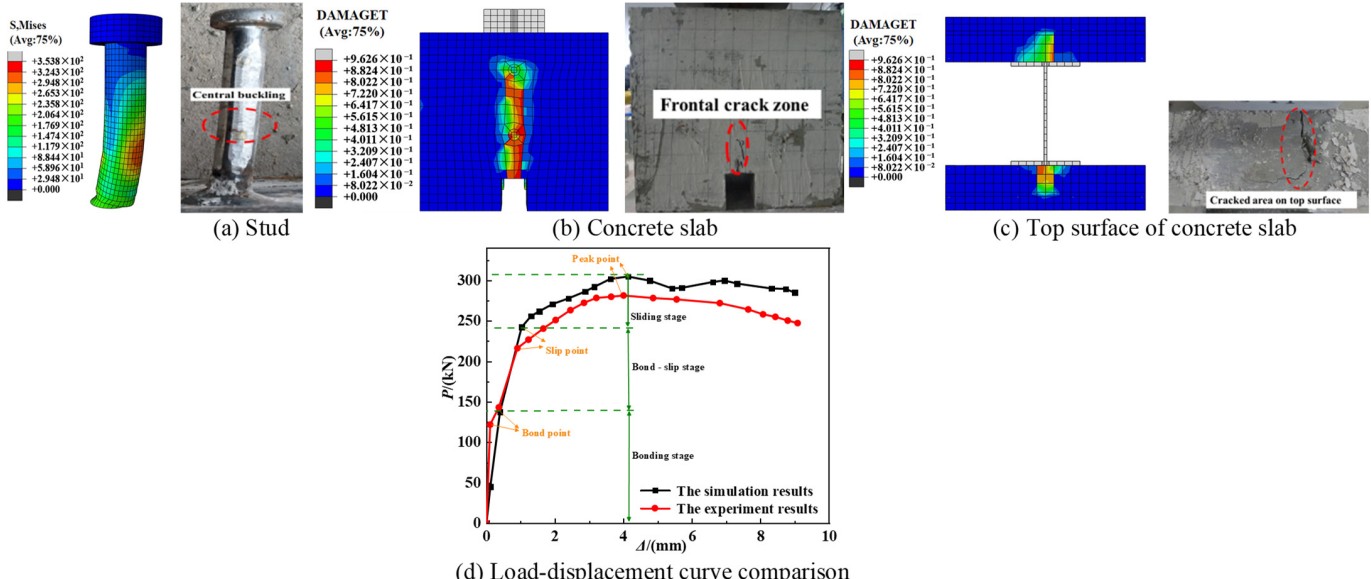

**Figure 9.** Comparison of stud test and simulation results.

The comparison between the simulation results of the triple-folded plate shear key and the experimental phenomena is shown in Figure 10a,d, and the comparison between the load-displacement curves is shown in Figure 11a. At the beginning of loading, the curves are in good agreement. After reaching the peak load, the bearing capacity decreases, and the trend is roughly the same. The ultimate bearing capacity of numerical calculation is slightly higher than the test results, but the general agreement is good. The simulation results of the shear key with tie steel bar are compared with the experimental phenomena, as shown in Figure 10b,e, and the load-displacement curve is compared, as shown in Figure 11b. The bearing capacity and ductility of the specimen can be improved by penetrating the steel bar at the opening. Obvious deformation occurs in the middle of the tie steel bar, as shown in Figure 10c, which is basically consistent with the test results. From the analysis of bond-slip development law, the bond, bond-slip, slip, and failure stages are in good agreement. In summary, the finite element model has high accuracy and can better realize the bond-slip performance of the new shear key.

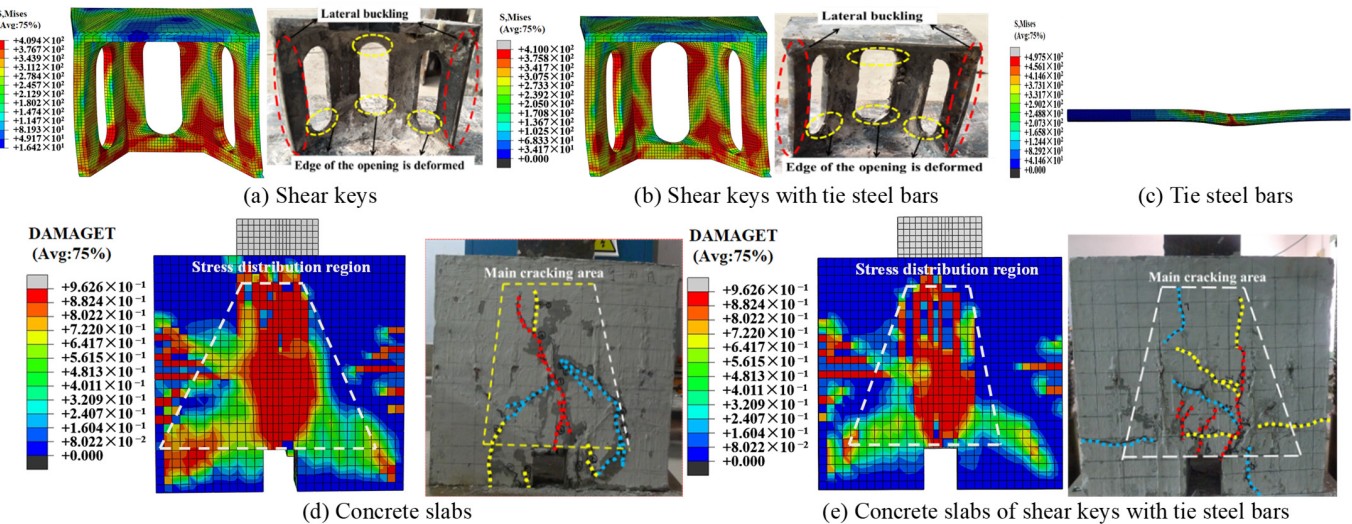

(a) Shear keys　　　(b) Shear keys with tie steel bars　　　(c) Tie steel bars

(d) Concrete slabs　　　(e) Concrete slabs of shear keys with tie steel bars

**Figure 10.** Comparison of shear key test and simulation results.

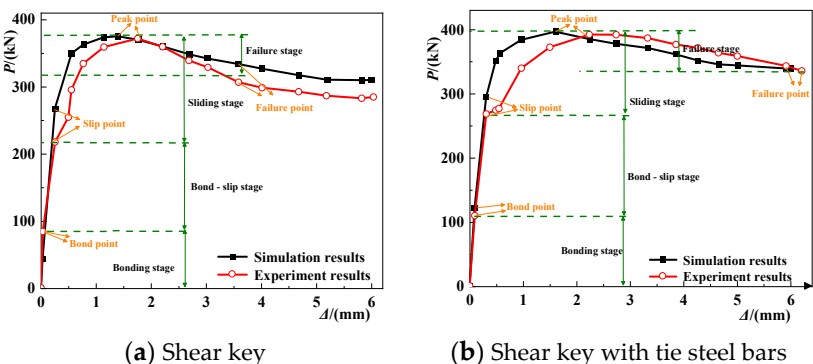

(**a**) Shear key　　　　　　(**b**) Shear key with tie steel bars

**Figure 11.** Comparison of load-displacement curves.

## 5. Finite Element Parametric Analysis

### 5.1. Web Height

The parameters of web height ($H$) are shown in Table 5. The comparison of load-displacement curves and parameter influence analysis are shown in Figure 12a. When the height is 70 mm, the bearing capacity is low, which is 299 kN; when the height is 150 mm, the bearing capacity of the shear key is the largest, reaching 409 kN. However, a minor increase in carrying capacity is compared with the verification model, about 9%. With the increase in height, the stiffness of shear keys is significantly different. When the web height is 90 mm, the stiffness is the largest; when it exceeds 110 mm, the stiffness is significantly reduced. When the web height is 90–130 mm, the ultimate load is basically the same, about 370 kN, and the model's bearing capacity decline is relatively mild, with good ductility. When the height increases to 150 mm and reaches the peak load, the bearing capacity drops sharply, and the ductility is poor. In summary, when the web height is 90 mm, the bearing capacity and stiffness of the shear key are high, the ductility is good, the constraint effect on the concrete slab is strong, and the integrity is good.

### 5.2. Opening Diameter

The opening diameter ($D$) parameters are shown in Table 5, and the comparison of load-displacement curves and parameter influence analysis are shown in Figure 12b. With the increase of opening diameter, the yield load decreases, and the stiffness decreases first and then increases; the opening diameter is 20 mm (minimum) or 40 mm (maximum), bearing capacity is low; when the hole diameter is 30 mm, the bearing capacity is the highest, reaching 389 kN. The hole diameter is 25 mm, 35mm; the curve drops gently,

and the ductility is good; the remaining model's curve reaches the peak load, the bearing capacity drops sharply, and the ductility is poor. In summary, when the opening diameter is too small, the shear key strength does not match the concrete strength, which affects the embedded effect between the shear key and the concrete and leads to the crushing failure of the concrete. However, when the opening diameter is too large, and the stiffness of the steel plate is weakened too much, the mechanical performance of the shear key will be reduced. Therefore, when the opening diameter is 25 mm–30 mm, the embedded effect of the shear key is the best, and the ability to work with the concrete is stronger.

**Table 5.** The shear key structural parameters.

| Specimen Number | $B$ | $T$ | $H$ | $D$ | $D_r$ | Specimen Number | $B$ | $T$ | $H$ | $D$ | $D_r$ |
|---|---|---|---|---|---|---|---|---|---|---|---|
| H70 | 60 | 6 | 70 | 25 | - | T10 | 60 | 10 | 90 | 25 | - |
| H90 | 60 | 6 | 90 | 25 | - | T12 | 60 | 12 | 90 | 25 | - |
| H110 | 60 | 6 | 110 | 25 | - | $D_r$10 | 60 | 6 | 90 | 25 | 10 |
| H130 | 60 | 6 | 130 | 25 | - | $D_r$12 | 60 | 6 | 90 | 25 | 12 |
| H150 | 60 | 6 | 150 | 25 | - | $D_r$14 | 60 | 6 | 90 | 25 | 14 |
| D20 | 60 | 6 | 90 | 20 | - | $D_r$16 | 60 | 6 | 90 | 25 | 16 |
| D25 | 60 | 6 | 90 | 25 | - | $D_r$18 | 60 | 6 | 90 | 25 | 18 |
| D30 | 60 | 6 | 90 | 30 | - | B50 | 50 | 6 | 90 | 25 | - |
| D35 | 60 | 6 | 90 | 35 | - | B60 | 60 | 6 | 90 | 25 | - |
| D40 | 60 | 6 | 90 | 40 | - | B70 | 70 | 6 | 90 | 25 | - |
| T4 | 60 | 4 | 90 | 25 | - | B80 | 80 | 6 | 90 | 25 | - |
| T6 | 60 | 6 | 90 | 25 | - | B90 | 90 | 6 | 90 | 25 | - |

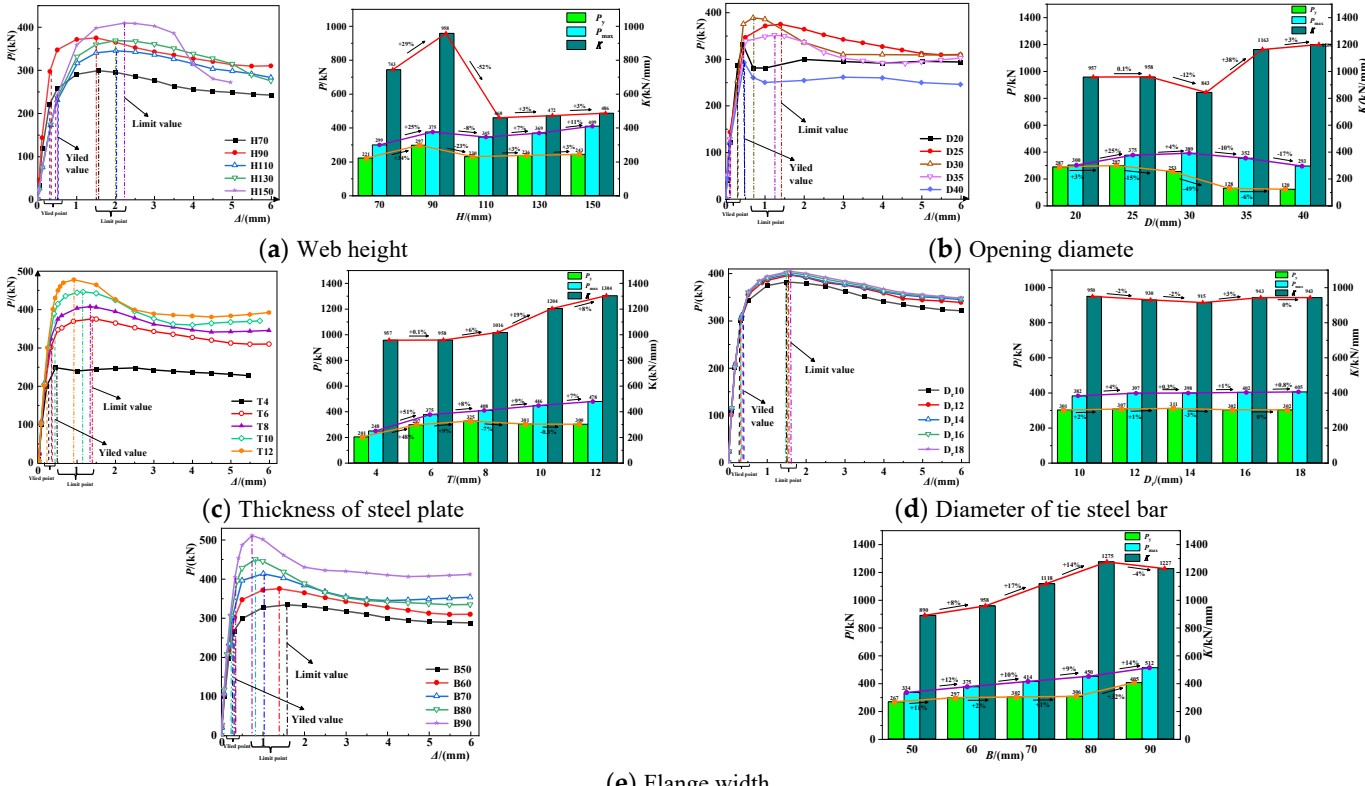

**Figure 12.** Load-displacement curve and parameter influence analysis.

### 5.3. Steel Plate Thickness

The parameters of steel plate thickness (*T*) are shown in Table 5, and the comparison of load-displacement curves and parameter influence analysis are shown in Figure 12c.

The stiffness and bearing capacity of the shear key increase with the thickness of the steel plate, but the yield load changes little; T = 4 mm, the lowest carrying capacity is 248 kN, T = 12mm, the highest carrying capacity is 478 kN; when the thickness of the steel plate increases from 4 mm to 6mm, the bearing capacity of the specimen increases by about 51%, and then with each increase of 2 mm, the bearing capacity increases by 8%, 9%, and 7%, respectively. The thickness of the steel plate is 4 mm–8 mm, the curve decreases gently, and the deformation capacity is good. When the thickness of the steel plate is 10 mm–12 mm, the bearing capacity decreases rapidly, and the deformation capacity is poor. In conclusion, when the thickness of the steel plate is 6 mm–8 mm, the bearing capacity is high, the ductility is good, and this is the recommended preference.

*5.4. Tie Steel Bar Diameter*

The diameter ($D_r$) of the tie steel bar is shown in Table 5, and the comparison of load-displacement curves and parameter influence analysis are shown in Figure 12d. With the increase of the diameter of the steel bar, the stiffness and ductility of the shear key are basically unchanged, while the ultimate bearing capacity increases, but the increase is small. For every 2 mm increase in the diameter, the bearing capacity increases by 4%, 0.3%, 1%, and 0.8%, respectively. The reason is that the difference between the stiffness and strength of the steel bar and the shear key is large. Overall, the tie steel bar has little effect on the shear key's bearing capacity but greatly influences the constraint capacity. If the steel bar is used to improve the mechanical performance of the shear key in engineering, the diameter of a 12 mm steel bar can obtain better results.

*5.5. Flange Width*

The flange width ($B$) parameter is detailed in Table 5, and the load-displacement curve and parameter influence analysis are shown in Figure 12e. With the increase of flange width, the stiffness and bearing capacity continue to increase, but when the flange width exceeds 80 mm, the stiffness decreases. When the width is 90 mm, the bearing capacity of the shear key is the largest, reaching 512 kN. Compared with the experimental model, the bearing capacity increases by about 37%. When the width increases from 80 mm to 90 mm, the bearing capacity of the shear key increases greatly, reaching 14%. When the width is 50 mm–70 mm, the curve decreases gently, and the ductility is good. When the width increases to 80 mm–90 mm, the bearing capacity decreases sharply, and the ductility is poor. In summary, the flange width greatly influences the bearing capacity. When the flange width is 60 mm–70 mm, the shear key has a high bearing capacity, good ductility, and strong constraint on concrete, so this is the recommended preference.

## 6. Bearing Capacity Calculation

The force analysis of shear keys is illustrated in Figure 13. The bearing capacity is mainly provided by the folded web ($V_1$), concrete tooth block ($V_2$), concrete at the corner ($V_3$), and flange plate ($V_4$). Due to the poor bond between steel and concrete, its role is ignored in the calculation. $V_1$: since deformation mainly occurs in the lateral web of the opening, the bearing capacity is determined by the yield strength and cross-sectional area of the steel plate; $V_2$: represents the biting force of the concrete block in the hole, which is determined by the contact surface area and the compressive strength of concrete; $V_3$: denotes the shear bearing capacity of the corner web, which is obtained by the compressive strength of the concrete in this part because of the large stiffness of the corner; $V_4$: is the shear capacity of the flange, as it remains elastic, and is determined by the compressive capacity of the concrete with which it is in contact.

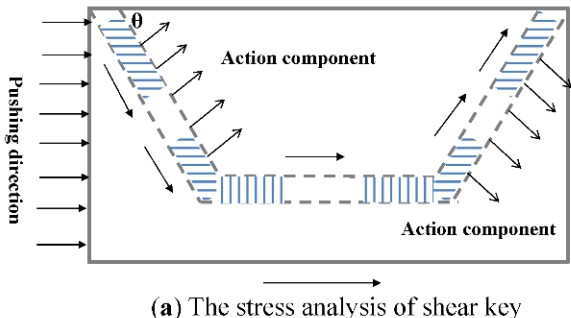

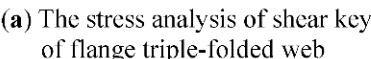

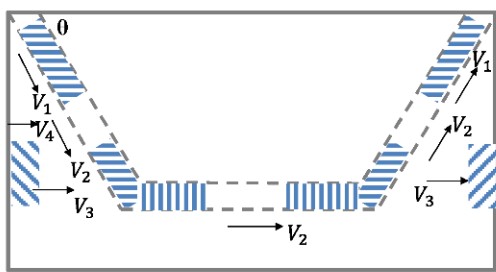

(**a**) The stress analysis of shear key of flange triple-folded web

(**b**) Simplified analysis of bearing capacity of the shear key of flange triple-folded web

**Figure 13.** Force analysis of the shear key.

According to the results of finite element parameter analysis, the influences of the web height, opening diameter, the thickness of the steel plate, the diameter of the tie steel bar, and flange width on the bearing capacity of shear keys are obtained. It is found that $V_1$ and $V_3$ contribute the most to the bearing capacity. $V_3$ is mainly affected by the flange width, opening diameter, and web height. $V_1$ is mainly affected by the flange width, the opening diameter, and the thickness of the steel plate. The contribution of $V_2$ and $V_4$ to bearing capacity is secondary and less affected by these factors, only affected by the thickness of the steel plate. According to the superposition principle, the shear bearing capacity of the shear key is calculated as follows:

$$V_{\mathrm{u}} = \gamma_{\mathrm{B}}\gamma_{\mathrm{D}}(\gamma_{\mathrm{T}}V_1 + \gamma_{\mathrm{H}}V_3) + \gamma_{\mathrm{T}}V_2 + \gamma_{\mathrm{T}}V_4 \tag{6}$$

where $V_{\mathrm{u}}$ is the ultimate bearing capacity of shear keys; $\gamma_{\mathrm{B}}$ is the influence coefficient of the flange width; $\gamma_{\mathrm{H}}$ is the influence coefficient of the web height; $\gamma_{\mathrm{D}}$ represents the influence coefficient of the embedded; and $\gamma_{\mathrm{T}}$ is the influence coefficient of the thickness of steel plate.

$$V_1 = nb_1 T f_{\mathrm{y}} \alpha_1 \beta \tag{7}$$

where $b_1$ is the width of the steel plate outside the opening; $T$ denotes the thickness of the web; $f_{\mathrm{y}}$ denotes the yield strength of steel; $\alpha_1$ is the yield strength reduction factor, which is 0.9 according to the test results; $n$ is the increase coefficient of web area, taking 1.25; and $\beta$ is the uneven strength reduction coefficient of the left and right folded plate, and the value is 2.0.

$$V_2 = H_1 T f_{\mathrm{ck}} \alpha_2 \beta \cos\theta \tag{8}$$

where $H_1$ is the opening height; $\theta$ is the angle of the folding plate; $f_{\mathrm{ck}}$ refers to the axial compressive strength of concrete; and $\alpha_2$ represents the strength reduction factor considering the influence of edge stress concentration, 0.85 is selected according to the test results.

$$V_3 = nH b_2 f_{\mathrm{ck}} \alpha_3 \beta \sin\theta \tag{9}$$

where $H$ is the height of the web; $b_2$ denotes the width of the inner web of the opening; and $\alpha_3$ is the reduction coefficient of the local compressive strength of concrete, which is 0.9 according to the test results.

$$V_4 = B T f_{\mathrm{ck}} \alpha_2 \tag{10}$$

where $B$ is the flange width.

The change of web height mainly affects the bearing capacity $V_3$, so the influence coefficient $\gamma_{\mathrm{H}}$ of web height is introduced, defined as the ratio of $V_{\mathrm{um}} - 2\gamma_{\mathrm{T}}(V_1 + V_2) - 2\gamma_{\mathrm{B}}\gamma_{\mathrm{D}}\gamma_{\mathrm{T}}V_4$ to $2V_3$. With the verification model as the standard, the value of $\gamma_{\mathrm{H}}$ is fitted and analyzed, as shown in Figure 14a. The formula is as follows:

$$\gamma_{\mathrm{H}} = 0.23 + 0.01762H - 1.0264 \times 10^{-4} H^2 \ (70 \le H \le 150) \tag{11}$$

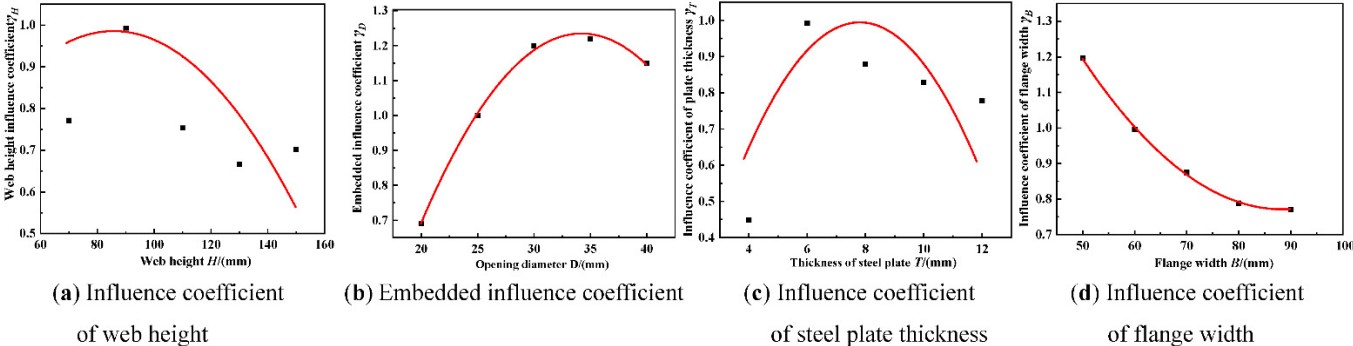

**(a)** Influence coefficient of web height    **(b)** Embedded influence coefficient    **(c)** Influence coefficient of steel plate thickness    **(d)** Influence coefficient of flange width

**Figure 14.** Fitting curves for influence coefficients.

The change of the hole diameter affects the overall embedded effect of the specimen, so the embedded influence coefficient $\gamma_D$ is defined as the ratio of $V_{um} - 2\gamma_T(V_2 + V_4)$ to $2\gamma_B(\gamma_T V_1 + \gamma_H V_3)$. As shown in Figure 14b, the calculation results are fitted. The calculation formula of the embedded influence coefficient $\gamma_D$ is as follows:

$$\gamma_D = -1.91486 + 0.18394D - 0.00269D^2 \left(20 \leq D \leq 40\right) \tag{12}$$

The thickness of steel plate mainly affects the bearing capacity $V_1$, $V_2$, and $V_4$, so the influence coefficient $\gamma_T$ of steel plate thickness is introduced, defined as the ratio of $V_{um} - 2\gamma_B\gamma_D\gamma_H V_3$ to $2(\gamma_B\gamma_D V_1 + V_2 + V_4)$. As shown in Figure 14c, the calculated values are fitted. The formula for calculating the influence coefficient of plate thickness is as follows:

$$\gamma_T = -0.461 + 0.373T - 0.023T^2 \left(4 \leq T \leq 12\right) \tag{13}$$

The influence coefficient of flange width $\gamma_B$ is defined as the ratio of $V_{um} - 2\gamma_T(V_2 + V_4)$ to $2\gamma_D(\gamma_T V_1 + \gamma_H V_3)$. Then, the calculated values are fitted, as shown in Figure 14d. The formula for calculating the influence coefficient of flange width is as follows:

$$\gamma_B = 3.01043 - 0.05061B + 2.85789 \times 10^{-4}B^2 \left(50 \leq B \leq 90\right) \tag{14}$$

It can be seen from Figure 14 that the fitting law of web height influence coefficient $\gamma_H$ and steel plate thickness influence coefficient $\gamma_T$ is not very ideal. The main reason is that with the decrease of shear key height, the deformation capacity decreases, resulting in the transformation of failure mode from shear key failure to concrete failure. At the same time, with the increase of the shear key's height, the top's deformation is larger and larger, and the influence of the bending moment is enhanced. The failure model of the shear key may change from shear failure to bending shear failure, which causes the fluctuation of the $\gamma_H$ fitting curve. When the steel plate is too thin, the stiffness of the shear key does not match that of the concrete, resulting in a significant decrease in performance, which causes a significant fluctuation of the $\gamma_T$ fitting curve. The calculated value of bearing capacity is compared with the simulated value, and the results are detailed in Table 6. The error between the two is basically within ±10%, and the accuracy is high, which can provide the theoretical basis for the practical engineering application of shear keys with triple webs.

**Table 6.** Comparison of ultimate bearing capacity between calculated and simulation values.

| Specimen | $V_u$ | $V_{um}$ | Error Value | Specimen | $V_u$ | $V_{um}$ | Error Value |
|---|---|---|---|---|---|---|---|
| H70 | 318 | 299 | 6.4% | T4 | 271 | 248 | 9.3% |
| H90 | 364 | 375 | −2.9% | T6 | 363 | 375 | −3.2% |
| H110 | 392 | 361 | 8.6% | T8 | 446 | 408 | 9.3% |
| H130 | 393 | 369 | 6.5% | T10 | 485 | 446 | 8.7% |
| H150 | 355 | 409 | −13.2% | T12 | 447 | 478 | −6.5% |

**Table 6.** *Cont.*

| Specimen | $V_{\mathrm{u}}$ | $V_{\mathrm{um}}$ | Error Value | Specimen | $V_{\mathrm{u}}$ | $V_{\mathrm{um}}$ | Error Value |
|---|---|---|---|---|---|---|---|
| D20 | 286 | 300 | −4.7% | B50 | 322 | 334 | −3.6% |
| D25 | 364 | 375 | −2.9% | B60 | 365 | 375 | −2.7% |
| D30 | 380 | 389 | −2.3% | B70 | 398 | 414 | −3.9% |
| D35 | 347 | 352 | −1.4% | B80 | 436 | 450 | −3.1% |
| D40 | 282 | 293 | −3.8% | B90 | 495 | 512 | −3.3% |

NOTE: $V_{\mathrm{u}}$ is the theoretical calculation value; $V_{\mathrm{um}}$ is the simulated value.

## 7. Bond-Slip Performance under Seismic Reciprocating Action

The bond-slip performance of shear keys under seismic action is quite different from that under unidirectional pushout, resulting in different degrees of degradation of bearing capacity, stiffness, and deformation capacity. Based on the static analysis test, the horizontal hysteresis test simulation was carried out for stud connectors and shear keys to analyze the two's bond-slip performance and failure mechanism under seismic action. The hysteresis analysis model was established according to the experimental model. The hysteretic constitutive model is adopted for concrete, and the bilinear kinematic hardening model is adopted for steel, as shown in Section 4.2.

### 7.1. Load-Displacement Hysteresis Curve

The load-displacement hysteresis curves of each model are shown in Figure 15. The hysteresis of the shear key is full, exhibiting a distinct shuttle-shape. The hysteresis curves of the stud are bow-shaped, and there is a certain pinching effect therein. After the shear keys reach the yield load, the stiffness degrades; after reaching the ultimate load, the bearing capacity decreases, and the hysteresis area increases. The bearing capacity of studs did not decrease. Through analysis of the hysteresis curves, it is found that the hysteresis characteristics of the shear key and the stud are pretty different. The former has a prominent degradation stage and a larger hysteresis area. As a flexible shear member, the stud has better deformation ability under earthquake load, but the consumption of seismic capacity is much lower than that of the new shear key.

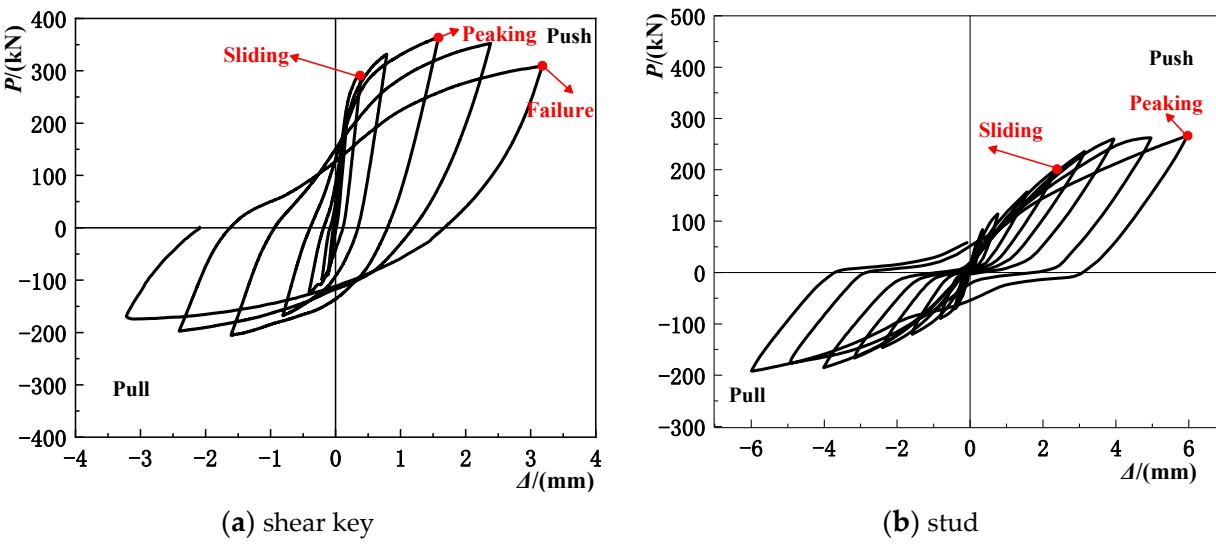

(**a**) shear key  (**b**) stud

**Figure 15.** Hysteresis curves.

### 7.2. Stress Distribution and Failure Characteristics

Under cyclic loading, the stress distribution of shear keys and concrete slabs is shown in Figure 16a,b. At the beginning of loading, the shear key is basically in an elastic state, the maximum stress is concentrated in the lower part of the web, and the stress distribution is relatively uniform. As the load continues, the shear key web begins to yield, and the

stress distribution range becomes larger; after stopping loading, the maximum stress is distributed at the edge of the opening, and the flange is basically in an elastic state, and the web completely yield. During the whole loading process, the maximum stress first appears near the shear key and at the bottom of the slab, which is distributed in a vertical stripe shape. As the load continues, the stress distribution appears in the 45° direction and the middle of the concrete; the concrete slab is seriously damaged after stopping loading. The maximum stress distribution range accounts for more than 60% of the slab surface. Under the action of horizontal hysteretic load, first, the roots of the stud yield, and as the load increases, the failure area develops upward until the middle of the stud. The concrete slab is first damaged at the top, and with the increase of load, the failure area develops downward, forming a strip plastic zone along the vertical direction. Next, some damage is formed horizontally, and the overall plastic region accounts for about 15%, as shown in Figure 17. Overall, the damage degree of the stud is light, only the root yield. The failure area of a concrete slab under stud constraint is significantly reduced compared with the failure area of the new shear key; the constraint capacity under earthquake is significantly lower than the new shear key.

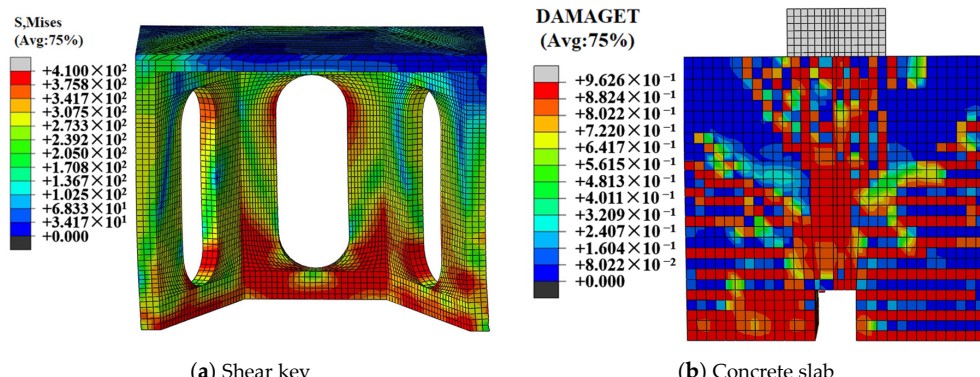

(**a**) Shear key                    (**b**) Concrete slab

**Figure 16.** Failure phenomenon of shear keys.

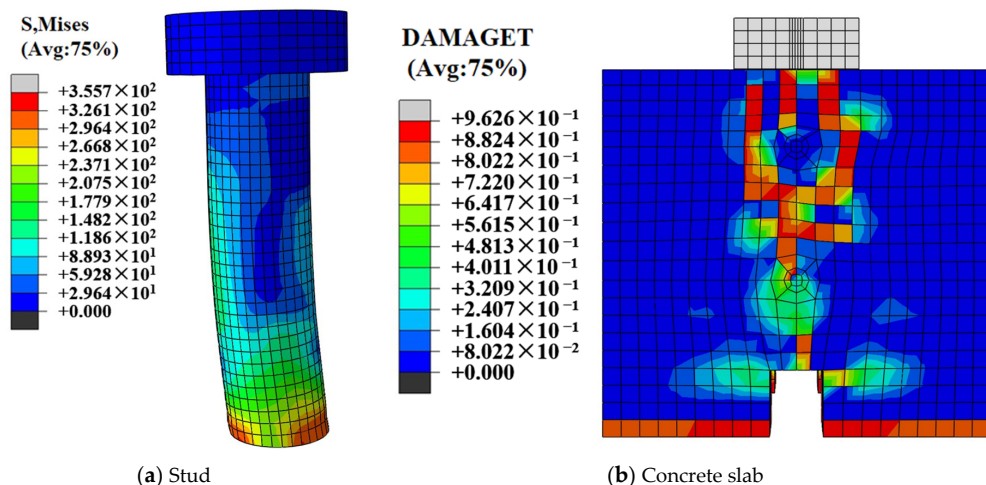

(**a**) Stud                    (**b**) Concrete slab

**Figure 17.** Failure phenomenon of studs.

*7.3. Bond-Slip Performance under Horizontal Push-Out and Hysteresis Action*

The comparison of load-displacement curves under monotonic and cyclic loading is shown in Figure 18. Under static push-out, the ultimate bearing capacities of shear keys and studs are 375 kN and 305 kN, respectively; under the repeated load, respectively, 366 kN and 264 kN, compared with the static load, the shear key bearing capacity decreased by 2.4%, the stud bearing capacity decreased by 13%. Under repeated load, the stiffness of the stud is significantly reduced, and the stiffness of the shear key is also degraded to a certain extent. The bond-slip performance under cyclic loading is analyzed. Due

to the large stiffness of the shear key, the concrete damage is more serious under cyclic loading. The stiffness of the stud as a flexible shear connector is slight, and its excessive deformation leads to the tensile failure of the root concrete. The bearing capacity also has 13% degradation. The triple-folded plate shear key has moderate stiffness and good embedment with concrete.

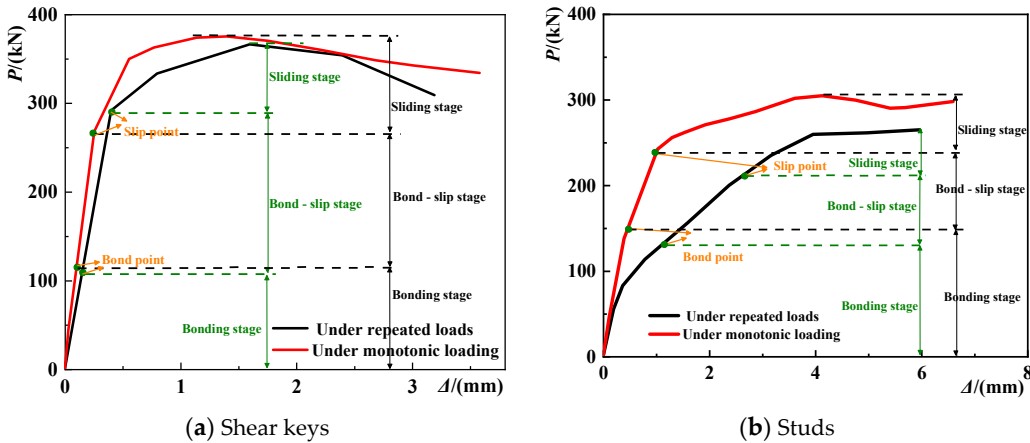

(**a**) Shear keys          (**b**) Studs

**Figure 18.** Comparison of load-displacement curves under monotonic and cyclic loading.

### 7.4. Skeleton Curves Analysis

The skeleton curves of shear keys and studs are shown in Figure 19. The yield point $P_y$ and the peak point $P_{max}$ (the yield displacement $\Delta_y$ is calculated by the equivalent elastic-plastic yield method, and the displacement $\Delta_{max}$ of the peak load is taken as the failure displacement $\Delta_u$) are shown in the figure. Overall, the shear key skeleton curve experienced three stages: bond, slip, and failure. The skeleton curve of studs only experienced two stages of bond and slip. Compared with studs, the shear key had larger initial stiffness and higher bearing capacity under repeated load, but its deformation was relatively poor.

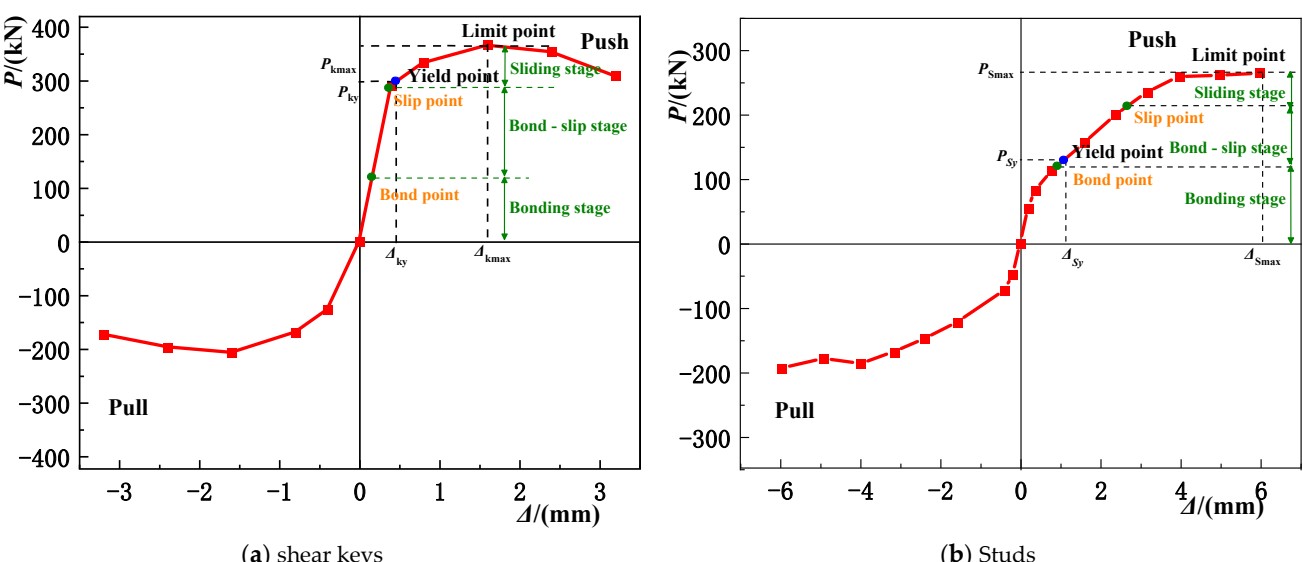

(**a**) shear keys          (**b**) Studs

**Figure 19.** Skeleton curves of specimens.

In this paper, the displacement ductility coefficient $\mu = \Delta_{max}/\Delta_y$ is used to measure the deformation capacity of the specimen. The calculated values of the displacement ductility coefficient of the shear key and the stud are shown in Table 7. Under repeated load, the ductility coefficient of the shear key is 3.3, and the ductility coefficient of the stud is 5.4. The ductility coefficient of the stud is about 1.6 times that of the shear key, and the

deformation capacity is strong. However, the ductility coefficient of the shear key is greater than 3, and it also shows good deformation ability.

**Table 7.** Displacement ductility coefficient.

| Specimen | Bond-Slip Load $P_S$/(kN) | Displacement of Bond-Slip $\Delta_S$/(mm) | Yield Load $P_y$/(kN) | Displacement of Yield $\Delta_y$/(mm) | Peak Load $P_{max}$/(kN) | Displacement of Peak $\Delta_{max}$/(mm) | Ductility Coefficient $\mu$ |
|---|---|---|---|---|---|---|---|
| shear key | 296 | 0.43 | 298 | 0.48 | 366 | 1.59 | 3.30 |
| Stud | 211 | 2.64 | 138 | 1.10 | 264 | 5.98 | 5.40 |

*7.5. Stiffness Degradation and Energy Dissipation Capacity Analysis*

The comparison of the secant stiffness *K* of each model is shown in Figure 20. Under the action of horizontal push-out, the initial stiffness of the triple-folded web shear key is much larger than that of the stud, by about four times. Large initial stiffness can better control deformation; subsequently, the stiffness is degraded and gradually reduced to 0. In the whole process, the internal force can be well adjusted. Overall, the stiffness of the new shear key changes greatly, which is very beneficial to controlling slip deformation and internal force transfer. The equivalent viscous damping coefficient $h_e$ measures the energy dissipation capacity of the model. The change of the equivalent viscous damping coefficient of the shear key and the stud is shown in Figure 21. Overall, the damping coefficient of the shear key and the stud increases with the displacement. The maximum damping coefficient of the shear key is 0.26, and the stud is 0.16. Compared with the stud, the damping coefficient of the shear key is increased by 1.6 times. The seismic energy dissipation capacity is strong, and the advantage is obvious.

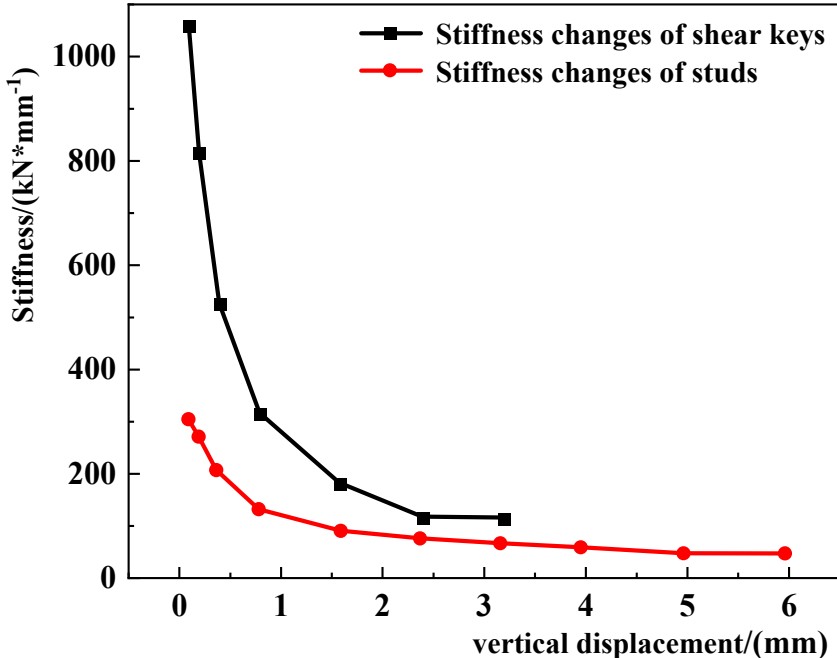

**Figure 20.** Curve of stiffness change of specimens.

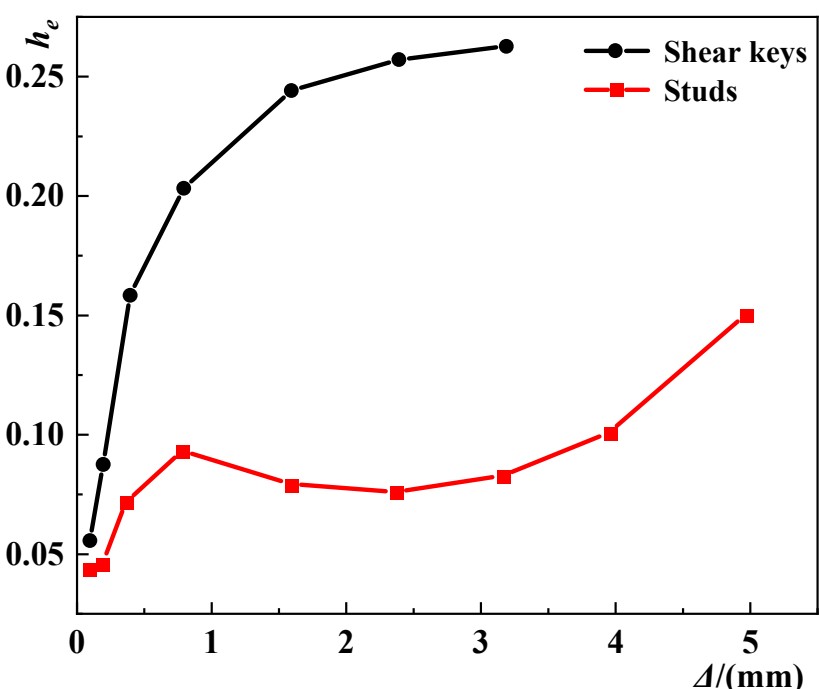

**Figure 21.** Equivalent viscous damping coefficient of specimens.

## 8. Conclusions

In this paper, the push-out test, numerical analysis, and bearing capacity calculation are carried out for the bond-slip performance of the shear key. The main conclusions are as follows:

(1) The push-out test shows that the new shear key has high bearing capacity. Compared with the stud connector, the sliding load increases by more than 45%, and the ultimate load increases by more than 32%. The safety margin is slightly small. At the same time, it has good deformation ability and basically realizes the design purpose of "strength" and "deformation" coordination; the failure mode of the shear key is as follows: bending occurs along the push-out direction, and the deformation of both sides of the web and the edge of the opening is serious. The concrete slab forms a trapezoidal cracking area centered on the shear key. Compared with the stud's point constraint, the shear key's constraint range is wider, and the integrity is better. In addition, the study on the influence of tie steel bars shows that it has little effect on the bearing capacity of the shear key. However, it can increase the constraint capacity and ductility of the shear key to a certain extent.

(2) Through parameter analysis, it is found that the bearing capacity of triple-folded shear key increases with the increase of web thickness, flange width, and diameter of penetrating steel bar and decreases with the decrease of opening diameter in a certain range; considering the stiffness and ductility performance, the best matching principle of the shear key structure parameters of the flange triple-folded web is obtained: flange width 60 mm–70 mm, plate thickness 6 mm–8 mm, web height 90 mm, and opening diameter 25 mm–30 mm; if the steel bar is inserted at the opening, the steel bar with a diameter of 12 mm should be preferred.

(3) According to the experimental results and numerical simulation, it is found that the width of the flange, the height of the web, the diameter of the opening, and the thickness of the steel plate have a great influence on the bearing capacity of shear keys. Through the fitting analysis of the numerical calculation results, the influence coefficient of web height $\gamma_H$, the embedded influence coefficient $\gamma_D$, the influence coefficient of steel plate thickness $\gamma_T$, and the influence coefficient of flange width $\gamma_B$ are obtained, and the calculation formula of ultimate bearing capacity is proposed.

(4) The study of seismic performance found that the load-displacement hysteretic curve of the shear key is full, showing good seismic performance. The ductility coefficient reaches 3.3, and the equivalent viscous damping coefficient is 0.26. The energy dissipation capacity is more than 1.6 times higher than that of the stud, and the stiffness can be more than four times higher than that of the stud. At the same time, the seismic bearing capacity of the shear key is less reduced than that of the unidirectional pushout strength, showing good comprehensive performance.

**Author Contributions:** Conceptualization, Z.W., Y.L., and H.Q.; methodology, Z.W. and H.Q.; software, Z.W.; validation, Z.W., Y.L., and H.Q.; investigation, H.Q. and Y.Y.; resources, Z.W.; data curation, Y.Y.; writing—original draft preparation, Z.W. and H.Q.; writing—review and editing, Y.L., H.G. and H.W.; visualization, Z.W. and H.Q.; supervision, Z.W., H.G., Y.L., and H.W.; project administration, Z.W.; funding acquisition, Z.W. All authors have read and agreed to the published version of the manuscript.

**Funding:** Supported by the National Natural Science Foundation of China (51978571) and the Key R&D Program of Shaanxi Province (2022SF-199, 2022SF-121).

**Institutional Review Board Statement:** Not applicable.

**Informed Consent Statement:** Not applicable.

**Data Availability Statement:** Data are contained within the article.

**Acknowledgments:** This work was supported by the National Natural Science Foundation of China and the Key R&D Program of Shaanxi Province.

**Conflicts of Interest:** The authors declare no conflict of interest. The company had no role in the design of the study; in the collection, analyses, or interpretation of data; in the writing of the manuscript, or in the decision to publish the results.

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
