# Peer review of "Push-Out Test and Hysteretic Performance Study of Semi-Rigid Shear Keys with the Triple-Folded Web of Flange"

_buildings, doi:10.3390/buildings12070991_

Round 1
Reviewer 1 Report
The manuscript entitled "Push-out test and hysteretic performance study of semi-rigid shear keys with the triple-folded web of flange” proposed a new type of flange triple web shear key and obtains the mechanical properties and failure mechanism of the new shear key through the push-out test. Moreover, a FE model was proposed and validated to conduct a parametric study.
The manuscript investigates a valuable topic in composite structures. However, the following comments should be addressed before any further processing.
Technical comments:
1- The technical writing and state of art of this manuscript are not good and should be improved.
2- Abbreviations should be defined at the first call in the manuscript and then they can be used throughout the manuscript.
3- The literature should be expanded to include the following references:
https://doi.org/10.1061/(ASCE)ST.1943-541X.0002326
4- Lines 106-107: What are the dimensions (diameter and height) of these two studs? This equivalence should be highlighted in the revised manuscript to make the provided comparison more realistic.
5- Figure 1: The dimensions are not clear. This figure should be improved.
6- Section 2.1: Typically, the steel beam is split into two halves until pouring the concrete slabs, and then they are welded together to conduct the push-out test. Did the authors use the same sequence? It is not easy to fabricate the two concrete slabs at the same time.
7- Lines 113-114: It is an unclear sentence. The authors should be clear to weld the shear keys on the top and bottom steel flanges. Also, how to assemble the precast unit should be clear.
8- The longitudinal and transverse reinforcement should be illustrated in Figure 2.
9- This number of shear keys represents what degree of shear connection? Is it a full or partial shear connection?
10- Section 2.2: How did the authors get these properties and according to which standards?
11- Table 2: What is the difference between MPa and N/mm2? They should be consistent throughout the manuscript.
12- Section 2.3: What was the loading rate (mm/minute)? What were the instrumentations? Did you measure the slip between the steel beam and concrete? What about strains? All of this information should be reported in the manuscript, not just shown on a figure.
13- The font of the text on the provided figures should be enlarged to be readable.
14- This reviewer highly recommends combining section 3.1 and section 3.2 together.
15- Figure 5: The caption of this figure is general. Which specimens are for this mode of failure?
16- Section 3.3: I think it will be better to change the title of this section to "Load-slip" or "Load-slippage" curves.
17- From Figure 6, looks like the typical stud shear connectors have better ductility than the proposed shear key. Higher stiffness and stronger capacity are not the only parameters that should be investigated.
18- Section 4.1: More features about the used elements should be provided like degrees of freedom and capability of deformation. Also, the reason for using these elements should be highlighted.
19- Lines 232-233: Unclear sentence. The authors should add "to simulate the contact between" or "to simulate the bond between".
20- Section 4.1: More information about the simulation of the slippage between the steel beam and concrete should be highlighted.
21- Section 4.1: More information about the element size, convergence issue, and the type of loading (force or displacement control) should be provided in this section.
22- Section 5: The symbol "H" should be highlighted in Figure 1. The same for the other parameters should be highlighted in their meanings in Figure 1.
23- Simulation of seismic action should be highlighted in the Finite Element model. The authors did not provide enough information about this simulation.
Author Response
Response to Reviewer 1 Comments
Dear Editor and Reviewers,
Firstly, we have much appreciated the reviewers for his/her constructive suggestions and comments. According to the comments of Reviewer#1 for the manuscript (ID: buildings-1788316 entitled “Push-out test and hysteretic performance study of semi-rigid shear keys with the triple-folded web of flange”), we have made the corresponding modification and clarification point by point. The changed contents are marked with red color.
Some statements and explanations about the revision in terms of Reviewer #1 are attached as follows.
Thank you for your attention.
Best regards,
Yours sincerely
Zhenshan Wang
Reviewer #1
Comment-1: The technical writing and state of art of this manuscript are not good and should be improved.
Response 1: Thanks to the reviewers for their valuable comments, we have improved the writing, and the language level has been substantially improved. For specific revisions, see the yellow highlighted part of the text.
Comment-2: Abbreviations should be defined at the first call in the manuscript and then they can be used throughout the manuscript.
Response 2: According to expert opinion, abbreviations have been defined at the first call. See the red part of the text for details.
Comment-3: The literature should be expanded to include the following references:
https://doi.org/10.1061/(ASCE)ST.1943-541X.0002326.
Response 3: Agreeing with the expert suggestion, we added the literature and modified the DOI format, see the red part in the text.
Comment-4: Lines 106-107: What are the dimensions (diameter and height) of these two studs? This equivalence should be highlighted in the revised manuscript to make the provided comparison more realistic.
Response 4: The studs are 90mm high and 22mm in diameter. According to experts' opinions, we have marked the stud's dimensions (diameter and height) in the text and Figure 1, as shown in red.
Comment-5: Figure 1: The dimensions are not clear. This figure should be improved.
Response 5: We are sorry for the unclear dimensions in the picture. Now Figure 1 has been modified to make the picture clearer.
Comment-6: Section 2.1: Typically, the steel beam is split into two halves until pouring the concrete slabs, and then they are welded together to conduct the push-out test. Did the authors use the same sequence? It is not easy to fabricate the two concrete slabs at the same time.
Response 6: The application of prefabricated composite floor slabs is increasing gradually in engineering. Given this situation, this paper made two floor slabs respectively for assembling to study the bond-slip performance of shear key and the constraint ability of concrete slabs in this form. The production process of specimens is shown in Figure 2. Although this approach is complicated, it is closer to the actual engineering situation of the assembled composite floor slab, so this paper adopts the processing form of assembling two floor slabs and then pouring concrete.
Comment-7: Lines 113-114: It is an unclear sentence. The authors should be clear to weld the shear keys on the top and bottom steel flanges. Also, how to assemble the precast unit should be clear.
Response 7: The shear key is welded to the upper flange of the steel beam. According to the expert opinion, the language has been modified to make the sentence more fluent and clear. See the red part in the text. See Section 2.1 and Figure 2 for the processing process of specimens.
Comment-8: The longitudinal and transverse reinforcement should be illustrated in Figure 2.
Response 8: Thanks to the advice given by the experts, the longitudinal and transverse reinforcement have been marked in Figure 2.
Comment-9: This number of shear keys represents what degree of shear connection? Is it a full or partial shear connection?
Response 9: The number of shear keys represents the degree of shear connection. In this paper, the bond-slip properties of the new shear key are studied. From the perspective of the research purpose, the primary purpose is to improve the mechanical performance and restraint capacity of shear connectors and to find a better balance between stiffness and deformation. Compared with studs, the performance of this new type of shear key has been dramatically improved. However, in terms of the overall number of settings, it is less than that of studs, so it is not easy to achieve a complete shear connection, and it should be a partial shear connection.
Comment-10: Section 2.2: How did the authors get these properties and according to which standards?
Response 10: In section 2.2, the concrete material property test is conducted according to Standard for Test Methods of Physical and Mechanical Properties of Concrete (GB/T50081-2019), and cubic blocks of 150mm×150mm×150mm determine concrete strength. The elastic modulus of concrete was tested with a prism block of 150mm×150mm×300mm. The steel plate and steel bar were sampled from the section steel, shear key, and the steel bar in the plate for testing. The tensile test was conducted according to Tensile test Method for Metal Materials at Room temperature (GB/T228.1-2010).
Comment-11: Table 2: What is the difference between MPa and N/mm2? They should be consistent throughout the manuscript.
Response 11: Thanks for the valuable advice given by the experts. MPa and N/mm2 in Table 2 are the same. We have corrected the mistakes in the paper, as shown in red.
Comment-12: Section 2.3: What was the loading rate (mm/minute)? What were the instrumentations? Did you measure the slip between the steel beam and concrete? What about strains? All of this information should be reported in the manuscript, not just shown on a figure.
Response 12: A long column press was used in the test device, as shown in Fig. 3. The loading rate was 0.15kN/s when loading was controlled by force, and each load of 100kN was held for two minutes. Under displacement control, the loading rate is 0.03mm/min. A measuring point D was arranged on the steel beam's web to measure the steel beam's slip, as shown in Fig. 3 (c). The pressure was obtained by the pressure sensor of the long column press. Based on expert advice, this information has been added to section 2.3, which is highlighted in red.
Comment-13: The font of the text on the provided figures should be enlarged to be readable.
Response 13: The text in the picture has been enlarged according to expert advice.
Comment-14: This reviewer highly recommends combining section 3.1 and section 3.2 together.
Response 14: In agreement with expert recommendations, sections 3.1 and 3.2 have been combined.
Comment-15: Figure 5: The caption of this figure is general. Which specimens are for this mode of failure?
Response 15: The common flange triple-folded web shear key applies to the failure mode shown in Fig. 5. In order to express the point of view in Fig. 5 more clearly, we have modified the title. See the part marked in red.
Comment-16: Section 3.3: I think it will be better to change the title of this section to "Load-slip" or "Load-slippage" curves.
Response 16: In agreement with expert advice, the title of Section 3.2 has been changed to load-slippage curves.
Comment-17: From Figure 6, looks like the typical stud shear connectors have better ductility than the proposed shear key. Higher stiffness and stronger capacity are not the only parameters that should be investigated.
Response 17: As a flexible connector, the stud has a higher deformation capacity, but its stiffness is smaller, and the number of arrangements is more lager. For structures with high shear bearing capacity requirements, too many shear keys may cause some inconvenience to construction. Therefore, a new type of shear key is proposed in this paper, which not only has greater rigidity but also has better deformation ability and improves the structure's integrity.
Comment-18: Section 4.1: More features about the used elements should be provided like degrees of freedom and capability of deformation. Also, the reason for using these elements should be highlighted.
Response 18: Thanks for the valuable comments from the experts. The solid element is suitable for large deformation analysis, and the calculation accuracy is high. When the mesh has bending deformation, the analysis accuracy will not be significantly affected. Therefore, the solid element C3D8R (8 Nodal hexahedra linear reduced-integration element). The slenderness of the steel bar is relatively large, so it is bending, shearing and torsion are ignored, and only the tensile force is considered, so the truss element T3D2 (2-node linear element) is used. The bottom of the concrete slab adopts a completely fixed boundary condition (U1=U2=U3=UR1=UR2=UR3=0). In order to avoid the buckling of the steel beam during the loading process, the displacement/rotation angle in the X and Y directions of the steel beam is set as 0 (U1=0, U2=0). Select the reference point 1 (RP-1) at a distance of 50mm from the center point of the top surface of the steel beam as the control point of the coupling constraint, and apply a displacement load of 6mm in the positive direction of Z. These details have been supplemented in the text and are highlighted in red.
Comment-19: Lines 232-233: Unclear sentence. The authors should add "to simulate the contact between" or "to simulate the bond between".
Response 19: Agree with the expert suggestion. In order to make the sentence clearer, we have revised the sentences in lines 232-233. See the red part in the text.
Comment-20: Section 4.1: More information about the simulation of the slippage between the steel beam and concrete should be highlighted.
Response 20: Thanks for the expert advice. Due to the poor bonding performance between steel and concrete, when establishing the model, the tangential bonding behavior of the two is described by a penalty coefficient, the friction factor is set to 0.05, and the normal behavior is hard contact, allowing contact after separation. This information has been supplemented to Section 4.1. See the red section.
Comment-21: Section 4.1: More information about the element size, convergence issue, and the type of loading (force or displacement control) should be provided in this section.
Response 21: Agree with the expert suggestion. The overall unit size of the concrete slab and steel beam is 25 mm, and the unit size of the concrete tooth block is 3 mm; the shear key unit size is 5 mm, and the steel plate is divided into three layers according to the thickness direction, and the grid of key parts is encrypted. The bottom of the concrete slab is completely fixed boundary conditions (U1 = U2 = U3 = UR1 = UR2 = UR3 = 0); in order to avoid the buckling of the steel beam during loading, the displacement/rotation angle is set to 0 in the X direction, and Y direction (U1 = 0, U2 = 0); the reference point 1 (RP-1) 50 mm away from the top center of the steel beam is selected as the control point of the coupling constraint. The displacement load of 6 mm is applied to the positive direction of Z. In order to improve the convergence, when meshing, the grid quality is improved by adjusting the grid size and attribute. When the contact master-slave surface is selected, the surface with large stiffness and coarse mesh is selected as the main surface; when concrete tooth blocks contact with the shear key, choose "delete interference" to remove the gap between the two. Information on element size, convergence issues, and loading types has been supplemented in Section 4.1. See red in the text.
Comment-22: Section 5: The symbol "H" should be highlighted in Figure 1. The same for the other parameters should be highlighted in their meanings in Figure 1.
Response 22: Thanks to the experts' opinions, the meanings of the shear key construction parameters have been marked in Figure 1.
Comment-23: Simulation of seismic action should be highlighted in the Finite Element model. The authors did not provide enough information about this simulation.
Response 23: This paper uses a push-out test to study the bond-slip properties of the new shear key. The method is unidirectional static loading. Earthquakes, as a reciprocating action, can cause more damage to structures. Based on the static research, this paper uses the numerical method to analyze the change of bond-slip properties of shear keys under reciprocating load. In the material constitutive relationship, the hysteretic constitutive model is used for concrete, and the bilinear follow-up model is used for steel, as shown in Sections 4-2. Given the problem of insufficient information, the supplement is made in this chapter. For details, see the red part marked in the text.
Special thanks to you for your good comments.
We appreciate for Editors/Reviewer, warm work earnestly, and hope that the correction will meet with approval, Once again, thank you very much for your comments and suggestions.

Reviewer 2 Report
The paper proposes a new type of flange triple web shear key and obtains the mechanical properties and failure mechanism of thenew shear key through the push-out test. The article is well written and both the computational and experimental work is nicely presented.
Some minor points for consideration:
1) In line 1 please define the article as "research article".
2) In figures (like fig.2) please take care so that the explanatory text does not interfere with the pictures.
3) In line 229 please define what ABAQUS version you have used and try to expand chapter 4 with one or more paragraphs explaining the set-up in the Simulia Abaqus software.
Author Response
Response to Reviewer 2 Comments
Dear Editor and Reviewers,
Firstly, we have much appreciated the reviewers for his/her constructive suggestions and comments. According to the comments of Reviewer#2 for the manuscript (ID: buildings-1788316 entitled “Push-out test and hysteretic performance study of semi-rigid shear keys with the triple-folded web of flange”), we have made the corresponding modification and clarification point by point. The changed contents are marked with red color.
Some statements and explanations about the revision in terms of Reviewer #2 are attached as follows.
Thank you for your attention.
Best regards,
Yours sincerely
Zhenshan Wang
Reviewer #2
Comment-1: In line 1 please define the article as "research article".
Response 1: Thanks for the valuable advice from the experts. The first line of the article indicates that it is a research article. See the red part of the article.
Comment-2: In figures (like fig.2) please take care so that the explanatory text does not interfere with the pictures.
Response 2: According to expert opinions, the pictures in the text are improved to avoid the situation where the description text interferes with the pictures.
Comment-3: In line 229 please define what ABAQUS version you have used and try to expand chapter 4 with one or more paragraphs explaining the set-up in the Simulia Abaqus software.
Response 3: Thanks to the experts' suggestions, the numerical simulation uses ABAQUS version 6.14. For the problem of insufficient model information raised by the experts, it has been supplemented in Section 4.1, including constraints, element types, boundary conditions, and meshes. See the red part.
Special thanks to you for your good comments.
We appreciate for Editors/Reviewer, warm work earnestly, and hope that the correction will meet with approval, Once again, thank you very much for your comments and suggestions.

Reviewer 3 Report
The authors presented very interesting studies of shear connectors in steel-concrete composite structures. Numerical tests and analyzes are carried out correctly, and the conclusions are supported by research results. Some specific comments are given below:
1. Page 4. Table 2 and Table 3. According to what standards, possibly on what samples, the mechanical parameters of the concrete specified in this table were determined? The same applies to the parameters of the steel.
2. Page 6. Fig.4. Referring to the photos of shear keys, after the examination, the question arises as to what the surfaces of these steel elements looked like before they were incorporated into the research models. Whether the steel was already surface corroded or not. This has an influence on the adhesion of steel to concrete and thus can influence the form of failure.
3. Page 14. Fig.14. The adopted dependencies describing the influence of coefficients in Figures (a) and (c) are incomprehensible. A broader comment from the authors on this issue is needed.
Author Response
Response to Reviewer 3 Comments
Dear Editor and Reviewers,
Firstly, we have much appreciated the reviewers for his/her constructive suggestions and comments. According to the comments of Reviewer#3 for the manuscript (ID: buildings-1788316 entitled “Push-out test and hysteretic performance study of semi-rigid shear keys with the triple-folded web of flange”), we have made the corresponding modification and clarification point by point. The changed contents are marked with red color.
Some statements and explanations about the revision in terms of Reviewer #3 are attached as follows.
Thank you for your attention.
Best regards,
Yours sincerely
Zhenshan Wang
Reviewer #3
Comment-1: Page 4. Table 2 and Table 3. According to what standards, possibly on what samples, the mechanical parameters of the concrete specified in this table were determined? The same applies to the parameters of the steel.
Response 1: Thanks to the experts for their valuable suggestions. The concrete material property test is conducted according to Standard for Test Methods of Physical and Mechanical Properties of Concrete (GB/T50081-2019), and cubic blocks of 150mm×150mm×150mm determine concrete strength. The elastic modulus of concrete was tested with a prism block of 150mm×150mm×300mm. The steel plate and steel bar were sampled from the section steel, shear key, and the steel bar in the plate for testing. The tensile test was conducted according to Tensile test Method for Metal Materials at Room temperature (GB/T228.1-2010). The obtained material performance indicators can lay the foundation for parametric analysis.
Comment-2: Page 6. Fig.4. Referring to the photos of shear keys, after the examination, the question arises as to what the surfaces of these steel elements looked like before they were incorporated into the research models. Whether the steel was already surface corroded or not. This has an influence on the adhesion of steel to concrete and thus can influence the form of failure.
Response 2: Thanks to the suggestions made by the experts, the surface of the shear key steel basically did not corrode before the concrete was poured; after the concrete was poured, the surface corrosion was not apparent because it was placed indoors and the conditions were good. Corrosion is common in engineering, and the effect of corrosion on shear bond-slip properties and failure modes will be further considered in subsequent studies.
Comment-3: Page 14. Fig.14. The adopted dependencies describing the influence of coefficients in Figures (a) and (c) are incomprehensible. A broader comment from the authors on this issue is needed.
Response 3: Thanks to experts' opinions, the fitting law of the influence coefficient of web height γH and the influence coefficient of plate thickness γT are not ideal. The main reason is that with the decrease of shear key height, the deformation capacity decreases, resulting in the failure mode from shear key failure to concrete failure. At the same time, as the height of shear keys increases, the deformation at the top becomes larger and larger, and the influence of the bending moment also increases. The failure model of shear keys may change from shear failure to bend-shear failure, resulting in the fluctuation of the curve in Fig. 14(a). When the steel plate is too thin, the stiffness of the shear key does not match that of the concrete, resulting in a significant decrease in performance, which causes the curve in Fig. 14(c) to fluctuate significantly. Given the reasons why the fitting curve is not ideal, the paper makes supplementary explanations. For details, see the red part marked in the paper.
Special thanks to you for your good comments.
We appreciate for Editors/Reviewer, warm work earnestly, and hope that the correction will meet with approval, Once again, thank you very much for your comments and suggestions.

Reviewer 4 Report
Dear Authors,
I have read manuscript titled: “Push-out test and hysteric performance study of semi-rigid keys with the triple-folded web of flange” with great attention.
In my opinion, the article has a good scientific level is well structured and can be published after MINOR REVISIONS, because it is an original and valuable work, but I have remarks to manuscript preparation and some concerns on the mechanical behaviour of the folded steel plate.
Some minor issues to be considered by the Authors:
Line 10: For the first use of abbreviation in the text the explanation has to be provided, e.g. PBL (perfobond leiste).
Line 69: The abbreviations (CFRP) has to be explained for the first use in the text.
Line 118-119: Please use the symbol of diameter or just wright "diameter", the letters are not acceptable.
Line 129-130: The same letters have to be used for the characteristics of the concrete, ether capital or small.
Table 3: The symbol of the diameter not the letter has to be used for denoting the diameter of the reinforcement.
Figure 3: The title reference has to be next to the figure.
Line 162-163: Incomprehensible statement, has to be rewritten.
Line 167-179: It is not clear what authors mean by the "phenomenon" in the titles of the figure 4. Has to be clarified.
Lien 208: The word "mild" is not a proper adjective to describe the decrease in bearing capacity, has to be rewritten.
Line 231: The description of the finite elements used for modeling the elements considered requires more detail specification, e.g. the degree of freedom, number of nodes, the deformability characteristics.
Line 246: What is the reason of the chosen value for the steel elastic modulus for reinforcement? The Table 3 shows the values of the elastic modulus for steel reinforcement varying from 183 to 189 GPa, not 214 GPa.
Line 283: Inexpedient use of words, the curve is not able to "load". The sentence has to be rewritten.
Fig. 9-10: In figure 9-10 the Authors indicate the buckling failure by the stress distribution. Stress concentration and the values indicate the yielding of the steel not the buckling failure. To simulate buckling behavior of the steel the appropriate analysis has to be used.
Line 393-394: The buckling of the steel pate cannot be determined by the yielding, these are two different mechanical characteristics of the steel element referring to the stiffness and strength, respectively. Based on the results and the slenderness of the plate it is more likely that the plate failed by the yielding not the buckling. The mechanics of the failure mechanism in steel plate has to be revised. Moreover it is recommended to consider the stress concentration due to the notch (opening).
Author Response
Response to Reviewer 4 Comments
Dear Editor and Reviewers,
Firstly, we have much appreciated the reviewers for his/her constructive suggestions and comments. According to the comments of Reviewer#4 for the manuscript (ID: buildings-1788316 entitled “Push-out test and hysteretic performance study of semi-rigid shear keys with the triple-folded web of flange”), we have made the corresponding modification and clarification point by point. The changed contents are marked with red color.
Some statements and explanations about the revision in terms of Reviewer #4 are attached as follows.
Thank you for your attention.
Best regards,
Yours sincerely
Zhenshan Wang
Reviewer #4
Comment-1: Line 10: For the first use of abbreviation in the text the explanation has to be provided, e.g. PBL (perfobond leiste).
Response 1: According to expert opinion, abbreviations have been defined at the first call. See the red part of the text for details.
Comment-2: Line 69: The abbreviations (CFRP) has to be explained for the first use in the text.
Response 2: According to expert opinion, abbreviations have been defined at the first call. See the red part of the text for details.
Comment-3: Line 118-119: Please use the symbol of diameter or just wright "diameter", the letters are not acceptable.
Response 3: Thanks to the valuable comments from experts, we have revised lines 118-119. See the red part in the text.
Comment-4: Line 129-130: The same letters have to be used for the characteristics of the concrete, ether capital or small.
Response 4: According to expert opinion, lines 129-130 have been modified. See the red part in the text.
Comment-5: Table 3: The symbol of the diameter not the letter has to be used for denoting the diameter of the reinforcement.
Response 5: The steel bar diameters in Table 3 have been represented by diameter symbols. See the red part of the text.
Comment-6: Figure 3: The title reference has to be next to the figure.
Response 6: The title has been placed directly below the figure based on expert opinion.
Comment-7: Line 162-163: Incomprehensible statement, has to be rewritten.
Response 7: Thanks to the suggestions from the experts, lines 162-163 have been rewritten. See the red part of the text.
Comment-8: Line 167-179: It is not clear what authors mean by the "phenomenon" in the titles of the figure 4. Has to be clarified.
Response 8: In Figure 4, the damage phenomena of the specimens refer to the test phenomena of shear keys and concrete slabs, and we have now modified the title of Figure 4 to avoid ambiguity. See the red marked part in the text.
Comment-9: Lien 208: The word "mild" is not a proper adjective to describe the decrease in bearing capacity, has to be rewritten.
Response 9: The word "mind" in the text is indeed inappropriate, and "decline mind" has been changed to "decline slow", see the red part in the text.
Comment-10: Line 231: The description of the finite elements used for modeling the elements considered requires more detail specification, e.g. the degree of freedom, number of nodes, the deformability characteristics.
Response 10: Thanks for the valuable comments from the experts. The solid element is suitable for large deformation analysis, and the calculation accuracy is high. When the mesh has bending deformation, the analysis accuracy will not be significantly affected. Therefore, the solid element C3D8R (8 Nodal hexahedra linear reduced-integration element). The slenderness of the steel bar is relatively large, so it is bending, shearing and torsion are ignored, and only the tensile force is considered, so the truss element T3D2 (2-node linear element) is used. The overall unit size of the concrete slab and section steel is 25mm, the concrete block unit is 3mm, the size of the shear key unit is 5mm, and the steel plate is divided into three layers according to the thickness direction. The mesh of key parts is densified. The bottom of the concrete slab adopts a completely fixed boundary condition (U1=U2=U3=UR1=UR2=UR3=0). In order to avoid the buckling of the steel beam during the loading process, the displacement/rotation angle in the X and Y directions of the steel beam is set as 0 (U1=0, U2=0). Select the reference point 1 (RP-1) at a distance of 50mm from the center point of the top surface of the steel beam as the control point of the coupling constraint, and apply a displacement load of 6mm in the positive direction of Z. These details have been supplemented in the text and are highlighted in red.
Comment-11: Line 246: What is the reason of the chosen value for the steel elastic modulus for reinforcement? The Table 3 shows the values of the elastic modulus for steel reinforcement varying from 183 to 189 GPa, not 214 GPa.
Response 11: Thanks to the expert's suggestion, the elastic modulus of the steel plate is 214 GPa, and the elastic modulus of the steel bar is 189 GPa. Due to our negligence, we did not express it clearly. Now that the elastic modulus of the steel bar has been added to the text. See the red part.
Comment-12: Line 283: Inexpedient use of words, the curve is not able to "load". The sentence has to be rewritten.
Response 12: Agree with the expert opinion. The sentence has been rewritten. See the red part in the text.
Comment-13: Fig. 9-10: In figure 9-10 the Authors indicate the buckling failure by the stress distribution. Stress concentration and the values indicate the yielding of the steel not the buckling failure. To simulate buckling behavior of the steel the appropriate analysis has to be used.
Response 13: Thanks to the suggestions made by the experts, due to the small size of the shear key, the relatively large thickness, and the relatively small width and thickness of the plate, the local stability effect is not significant. It is not very appropriate to use buckling failure in the paper. We have revised some of the content in the paper. See the red part.
Comment-14: Line 393-394: The buckling of the steel pate cannot be determined by the yielding, these are two different mechanical characteristics of the steel element referring to the stiffness and strength, respectively. Based on the results and the slenderness of the plate it is more likely that the plate failed by the yielding not the buckling. The mechanics of the failure mechanism in steel plate has to be revised. Moreover it is recommended to consider the stress concentration due to the notch (opening).
Response 14: Thanks to the suggestions made by the experts, the shear key has a small local buckling degree due to the influence of the size and thickness of the steel plate. However, due to the opening of the web, the outer part was subjected to reduced constraints, and certain bending deformation occurred in the test process. Considering the material strength under a complex stress state, the reduction coefficient of yield strength was introduced when calculating the bearing capacity of the outer steel plate, which was 0.9. As for the influence of stress concentration on the opening, due to the extensive length of the opening, the most significant stress concentration is located at the bottom and top of the hole, respectively. Through the test, it is found that the failure of the shear key is basically in the middle. Although the stress concentration greatly influences it, it is not in the failure section. Therefore, the paper is not considered.
Special thanks to you for your good comments.
We appreciate for Editors/Reviewer, warm work earnestly, and hope that the correction will meet with approval, Once again, thank you very much for your comments and suggestions.

Round 2
Reviewer 1 Report
The authors have addressed most of the reviewer's comments.